# A subset of CB002 xanthine analogs bypass p53-signaling to restore a p53 transcriptome and target an S-phase cell cycle checkpoint in tumors with mutated-p53

Liz Hernandez Borrero[1,2,3,4,5], David T Dicker[1,2,3,4,5], John Santiago[3], Jennifer Sanders[2,3,5,6], Xiaobing Tian[1,2,3,4,5], Nagib Ahsan[7], Avital Lev[2,3,4], Lanlan Zhou[1,2,3,4,5], Wafik S El-Deiry[1,2,3,4,5,8]*

[1]Laboratory of Translational Oncology and Experimental Cancer Therapeutics, The Warren Alpert Medical School, Brown University, Providence, United States; [2]The Joint Program in Cancer Biology, Brown University and the Lifespan Health System, Providence, United States; [3]Department of Pathology and Laboratory Medicine, The Warren Alpert Medical School, Brown University, Providence, United States; [4]Molecular Therapeutics Program, Fox Chase Cancer Center, Philadelphia, United States; [5]Cancer Center at Brown University, The Warren Alpert Medical School, Brown University, Providence, United States; [6]Department of Pediatrics, The Warren Alpert Medical School, Brown University, Providence, United States; [7]COBRE Center for Cancer Research Development, Proteomics Core Facility, Rhode Island Hospital, Providence, United States; [8]Hematology-Oncology Division, Department of Medicine, Rhode Island Hospital and Brown University, Providence, United States

*For correspondence: wafik@brown.edu

**Abstract** Mutations in TP53 occur commonly in the majority of human tumors and confer aggressive tumor phenotypes, including metastasis and therapy resistance. CB002 and structural-analogs restore p53 signaling in tumors with mutant-p53 but we find that unlike other xanthines such as caffeine, pentoxifylline, and theophylline, they do not deregulate the G2 checkpoint. Novel CB002-analogs induce pro-apoptotic Noxa protein in an ATF3/4-dependent manner, whereas caffeine, pentoxifylline, and theophylline do not. By contrast to caffeine, CB002-analogs target an S-phase checkpoint associated with increased p-RPA/RPA2, p-ATR, decreased Cyclin A, p-histone H3 expression, and downregulation of essential proteins in DNA-synthesis and DNA-repair. CB002-analog #4 enhances cell death, and decreases Ki-67 in patient-derived tumor-organoids without toxicity to normal human cells. Preliminary in vivo studies demonstrate anti-tumor efficacy in mice. Thus, a novel class of anti-cancer drugs shows the activation of p53 pathway signaling in tumors with mutated p53, and targets an S-phase checkpoint.

## Introduction

Tumor suppressor p53 responds to cell stress signals from DNA damage, oncogene activation, oxidative stress, and hypoxia. Upon activation by posttranslational modifications and oligomerization, p53 signals cell cycle arrest, apoptosis, or DNA repair, according to the extent of the cellular stress, thereby controlling cell fate and preventing tumorigenesis (*Riley et al., 2008*). Thus, it is not

surprising that *TP53* is the most commonly mutated gene (TCGA, 2020), including in ovarian, colorectal, esophageal, head and neck, lung, and pancreatic cancers that are the most affected sporadic human cancer types (*Olivier et al., 2010*). *TP53* is mutated in over 50% of human cancers and the other 50% involve a biological inactivation of its signaling pathway. Like other tumor suppressors, the mutated p53 protein results in loss-of-function but oligomerization can act in a dominant-negative fashion with regard to the remaining wild-type p53 allele. Unlike other tumor suppressors, mutant p53 protein can also acquire a gain-of-function which contributes to aggressive tumor phenotypes, including enhanced invasion, genomic instability, and therapy resistance (*Muller and Vousden, 2014*; *Dittmer et al., 1993*; *Lang et al., 2004*; *Xu et al., 2011*). Consequently, patients whose tumors carry p53 mutations have a poor prognosis and decreased overall survival (*Wattel et al., 1994*).

A common feature of cancer cells is genomic instability due to ineffective cell cycle checkpoint responses. Genomic instability is not necessarily due to defective checkpoints. The checkpoints may be intact but the repair may be deficient. Upon DNA damage, the normal cell cycle checkpoint response is to arrest the cell at the G1-phase. In cancer cells, the majority have an ineffective G1 checkpoint due to p53 mutation but retain a functional G2 checkpoint and thus have the ability to undergo cell arrest at the G2-phase. Cancer cells depend on bypassing intra-S-phase and G2/M checkpoints for unrestrained cell proliferation. Stress signal transduction in the p53 pathway is initiated by activation of kinases ataxia-telangiectasia-mutated (ATM), ataxia telangiectasia and Rad3 (ATR)-related, and downstream checkpoint kinases Chk1/2 that serve as signaling sensors and mediators of p53 activation. It has been a long-standing dogma that ATM/Chk2 and ATR/Chk1 are independently activated but recent studies provide evidence of cross-talk between the kinases (*Brown and Baltimore, 2003*; *Abraham, 2001*; *Smith et al., 2010*). Chk1/2 are kinases that participate in cell cycle checkpoint control, with Chk1 being active in S-phase and G2-phase, whereas Chk2 is active throughout the cell cycle (*Smith et al., 2010*; *Zhao and Piwnica-Worms, 2001*; *Chehab et al., 2000*).

Accumulation of genomic aberrations over time renders cancer cells vulnerable to checkpoint targeting therapy. Since the discovery of checkpoint targets, small molecule inhibitors have been pursued in combination with ionizing radiation and chemotherapy agents in order to deregulate checkpoints, thereby leading to cancer cell death. For example, combination of caffeine, a xanthine derivative, with irradiation or chemotherapy agents was found to deregulate the G2 checkpoint through ATM/ATR inhibition leading to therapy sensitization and enhanced cell death (*Russell et al., 1995*; *Sarkaria et al., 1999*). Nonetheless, translational cancer therapeutics studies were discontinued due to unachievable active concentrations in human plasma (*Lelo et al., 1986*). Thus, for the past two decades, the field has focused on the development of Chk1/2 inhibitors, which are in clinical trials (*Fracasso et al., 2011*; *Huang et al., 2012*; *Rogers et al., 2020*).

Another cancer therapeutic approach we and others have pursued involves restoration of p53 pathway signaling in tumors with mutant p53 or tumors that are null for p53. Despite efforts to restore the p53-pathway, to date, there are no FDA-approved drugs that functionally restore the p53 in tumors with mutated p53. We previously reported a p53-pathway restoring compound CB002 whose mechanism of action was not fully elucidated. We showed that CB002 leads to apoptotic cell death mediated by p53 target Noxa, a pro-apoptotic protein (*Hernandez-Borrero et al., 2018*). Here, we further evaluated more potent CB002-analog compounds and uncovered a unique mechanism of action suggestive of a novel class of anti-cancer drugs. Based on their molecular structure as xanthine derivatives, the novel class of CB002-analogs, unlike caffeine and other established xanthine derivatives, do not deregulate the G2 checkpoint. By contrast, the novel CB002-analog xanthines perturb S-phase and more importantly they restore the p53-pathway, a property not found with caffeine, pentoxifylline, and theophylline. We sought to characterize and define the new class of small molecules with anti-tumor properties by transcriptomic and proteomic analysis.

## Results

### CB002 and structural analogs restore the p53 pathway independently of p53, while xanthines such as caffeine, pentoxifylline, and theophylline do not

We sought to identify more potent analogs of parental xanthine compound CB002. We tested CB002-analogs in the ChemBridge library for the capability to induce the luciferase activity using a p53-regulated luciferase reporter stably expressed in the SW480 colorectal cancer cell line and also determined the IC50 values for the compounds by a CellTiter glow cytotoxicity assay (*Figure 1A–B*, *Figure 1—figure supplement 1*). The majority of the CB002-analogs tested, with the exception of analog #12, enhanced p53-reporter activity in a dose-dependent manner within a range of compound concentrations from 0 to 600 µM. We investigated the capability of a set of the CB002-analogs to induce apoptosis as indicated by Propidium Iodide (PI) staining sub-G1 population. As shown in *Figure 1C*, the treatment of tumor cells with CB002-analogs at an IC50 concentration (100 µM) resulted in a significant increase in sub-G1 content in SW480 cells. Moreover, the most potent CB002-analog #4 was found to increase cleaved-PARP and cytochrome C release from the mitochondria to the cytosol providing further evidence for apoptosis induction in SW480 tumor cells (*Figure 1D–E*, *Figure 1—figure supplement 2*). We investigated whether the p53-family member p73 may be a mediator of apoptosis and responsible for inducing p53 transcriptional targets by CB002-analogs.

### CB002 and structural analogs induce Noxa in an ATF3/4-dependent manner, independent of p73

As we previously showed for CB002 (19), p53-targets Noxa and DR5 were induced independently of p73 and PARP cleavage occurred despite effective p73 knockdown in CB002-analog #4 treated SW480 tumor cells (*Figure 1F*). Our previously published CB002 data indicated that Noxa plays a key role in mediating CB002-induced apoptosis (19). Thus, we sought to determine if CB002-analogs induce Noxa expression in four human colorectal cancer cell lines. In DLD-1 (p53$^{S241F}$), SW480 (p53$^{R273H,P309S}$), HCT116(p53$^{WT}$), and HCT116 p53$^{-/-}$ tumor cells expressing the exogenous R175H p53 mutant, Noxa protein expression was found to be induced, though some variation across cell lines was observed (*Figure 1G*). As these CB002-analogs are xanthine derivatives, we investigated whether other known xanthine derivatives, that is, caffeine, pentoxifylline, and theophylline can induce Noxa expression. However, we found that only the p53-pathway restoring CB002-analog xanthine compounds and not caffeine, pentoxifylline, and theophylline, induce Noxa protein expression (*Figure 1H*). Since Noxa can be transcriptionally activated independently of p53, we sought to explore other transcription factors involved in Noxa induction. We performed a knockdown of integrated stress response transcription factors ATF3/4 on SW480 cells. Knockdown of ATF3/4 upon treatment with 100 µM CB002 or 25 µM CB002-analog #4 abrogated Noxa protein induction (*Figure 1I*). Hence, our data suggests that ATF3/4 play a role in regulating Noxa expression.

### CB002-analog #4 treatment of human tumor cells enriches for cell cycle genes in addition to genes involved in the p53-pathway including apoptosis, indicating p53-pathway functional restoration

In order to understand how the CB002-analog molecules restore the p53-pathway, we performed a transcriptomic and proteomic analysis in SW480 cells treated with analog #4. Raw data from the transcriptomic and proteomic analysis can be found in *Supplementary file 1* and *Supplementary file 3*, respectively. To verify the quality of our transcriptomic data, the principal component (PC) plot was obtained. PC plots show that the factor with most variability within the samples was the difference between control and treatment (*Figure 2—figure supplement 1A–C*). Significant differentially expressed genes (DEGs) were defined by a false discovery rate (FDR)<0.05, and a total of 3362 genes met these criteria (*Figure 2—figure supplement 1D*). We then sought to identify the DEGs involved in the p53 pathway. To do this, a comprehensive known p53 target gene set used for comparison were the genes that have been previously shown to be directly regulated by p53 through chromatin immunoprecipitation assays assays and genes that were protein-coding genes in at least 3 of the 17 genome-wide data sets (from Fisher's analysis; *Fischer, 2017*). Out of

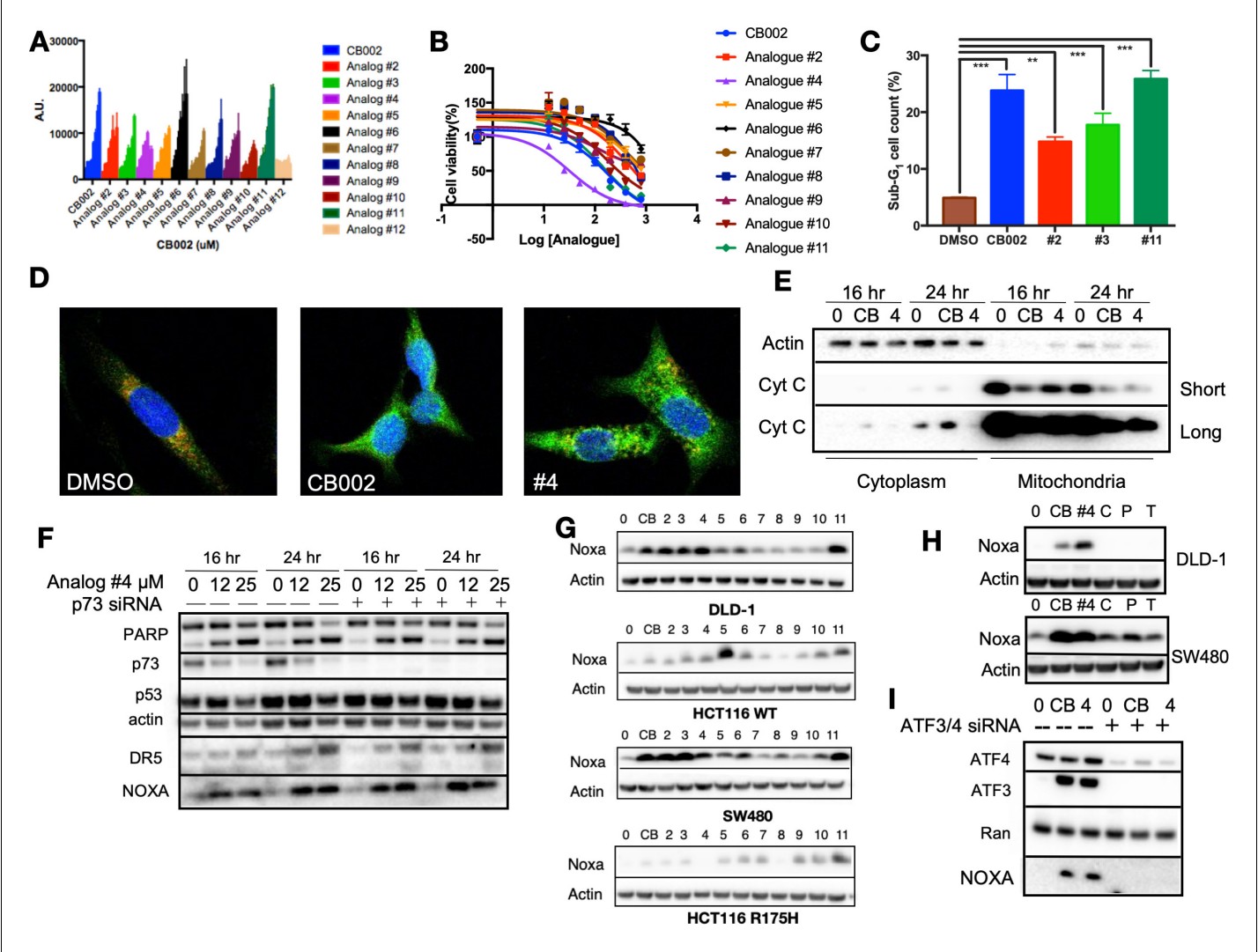

**Figure 1.** CB002 and structural analogs restore the p53 pathway, whereas other xanthines caffeine, pentoxifylline, and theophylline do not. CB002 structural analogs activate p53 reporter gene activity in SW480 cells in a dose-dependent manner (6 hr) (**A**). Therapeutic indices for CB002-structural analogs were determined in SW480 cells (48 hr) (**B**). Propidium iodide cell cycle analysis was performed to determine sub-G1 population at 48 hr of treatment with CB002-analogs at 100 µM in SW480 cells. Two-way ANOVA, p<0.05 (**C**). CB002-analog #4 restores the p53 pathway in SW480 cells, resulting in PARP cleavage independently of p73 (**D**). Immunofluorescence staining of Cyt-C (green), Tom20 (red) DAPI (blue) in SW480 treated as indicated for 48 hr (**E**). Noxa protein expression induced by CB002-analogs in DLD-1, SW480, HCT116, and HCT116 p53$^{(R175H)}$ colorectal cancer cells (24 hr) (**F**). p53-pathway restoring compounds have unique properties compared to other xanthine derivatives in their ability to induce Noxa expression, 24 hr treatment in DLD-1 cells (**G**). Xanthine derivatives CB002 and its analog induce Noxa expression but not caffeine, pentoxifylline, and theophylline at 24 hr in DLD-1 and SW480 cells (**H**). ATF3/4 mediate Noxa induction (**I**). Caffeine (C), Pentoxifylline (P), and Theophylline (T). Figures (**A**)–(**C**) were performed as three biological replicates. Experiments from figures (**D**)–(**I**) were performed at least twice and a representation of one is shown. The online version of this article includes the following figure supplement(s) for figure 1:

**Figure supplement 1.** CB002 and its analog #2–#11 chemical structures.
**Figure supplement 2.** CB002 and structural analog #4 induce apoptosis.

the 343 genes in the known p53 target gene set, 334 genes were tested in the microarray but only 197 genes met the low expression cutoff. From the 197 genes that met the low expression criteria, 102 genes were found to be differentially expressed (*Figure 2A*, *Supplementary file 2*). Gene ontology (GO) analysis of the 102 DEGs indicated that these genes are highly enriched in the regulation of programmed cell death (*Table 1*). A gene expression heatmap of these genes is shown in *Figure 2B*, and the majority of the genes are found to be upregulated by analog #4 treatment of tumor cells. We then performed a transcription factor analysis of all 3362 DEGs. Transcription factor

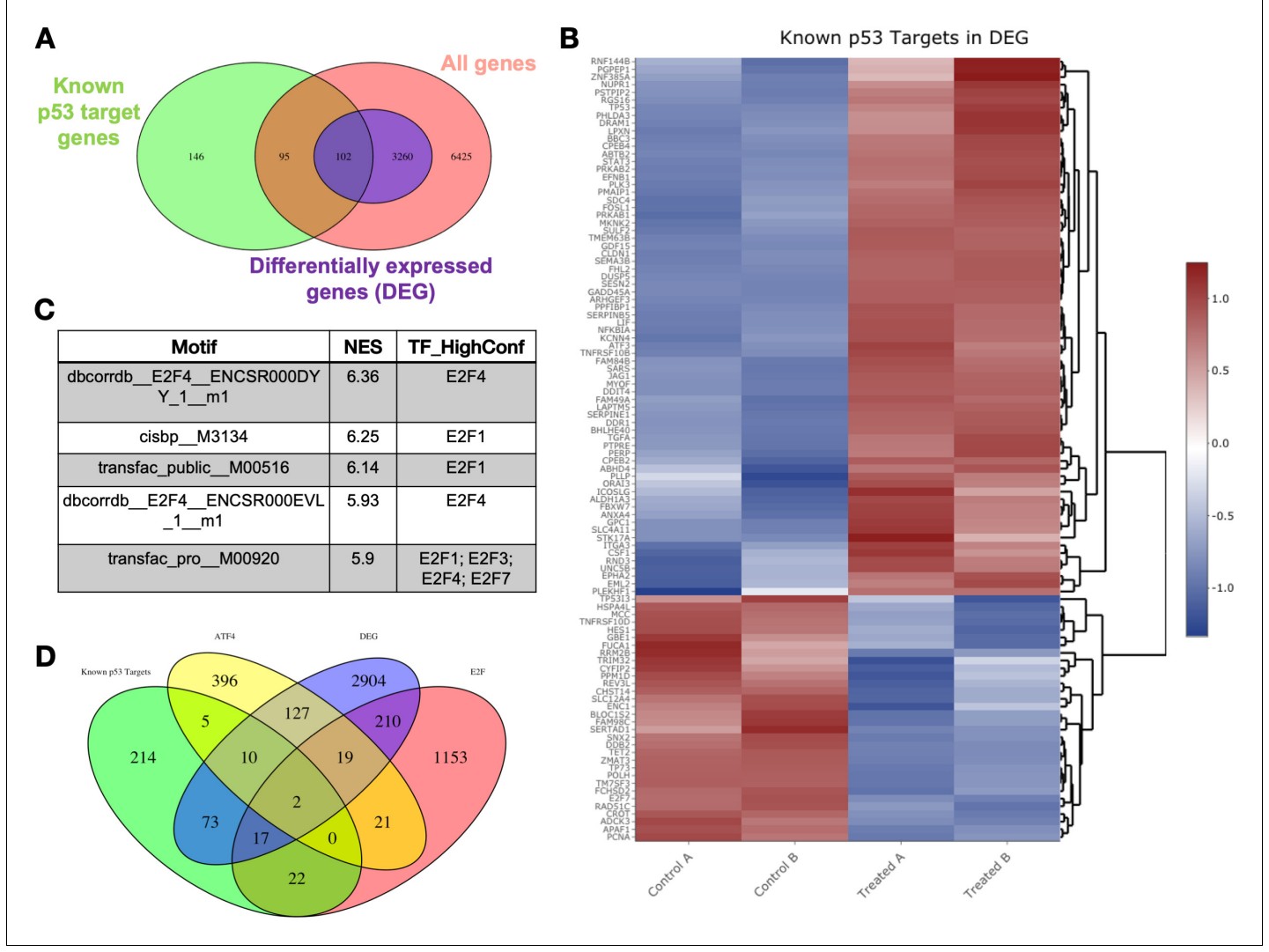

**Figure 2.** Transcriptomic pathway analysis of analog #4 reveals differentially expressed genes (DEGs) in tumor cells with mutant p53. SW480 cells were treated with analog #4 for 12 hr. Three-way Venn diagram of all genes tested that met the low expression cutoff (pink), DEGs with an FDR<0.05 (purple), and the known p53 target gene set (A). Heatmap of DEGs that overlapped with the known p53 target gene set (B). Predictive transcription factor analysis according to direct binding motif was performed for all the DEGs (total genes 3362) (C). Four-way Venn diagram of DEGs with an FDR<0.05 (purple), and the known p53 target gene set from *Table S3* of *Fischer, 2017* (green), ATF4 gene set (yellow), and E2F gene set (pink) (D). The online version of this article includes the following source data and figure supplement(s) for figure 2:

**Source data 1.** Gene expression values of DMSO vehicle control and analog #4 SW480 treated cells at 12 hr samples analysed by microarray Affymetrix Human Gene 2.0-ST array probe set.

**Source data 2.** Gene names from *Figure 2A–B* Venn diagram data sets containing all genes without the FDR of <0.05 filter, differentially expressed genes (DEG) with FDR of <0.05 filter and reference p53 data set from *Fischer, 2017*, Table S3.

**Figure supplement 1.** Transcriptomic analysis quality control principal component (PC) plots and false discovery rate (FDR) bar graph.

**Figure supplement 2.** Kyoto Encyclopedia of Genes and Genomes (KEGG) for the p53-pathway signaling.

**Figure supplement 3.** Heatmap of genes shown in *Figure 2—figure supplement 2* Kyoto Encyclopedia of Genes and Genomes (KEGG) p53-pathway signaling analysis.

**Figure supplement 4.** Kyoto Encyclopedia of Genes and Genomes (KEGG) for the cell cycle pathway.

**Figure supplement 5.** Heatmap of genes shown in *Figure 2—figure supplement 4* Kyoto Encyclopedia of Genes and Genomes analysis from the cell cycle pathway.

**Figure supplement 6.** Kyoto Encyclopedia of Genes and Genomes (KEGG) for the DNA replication pathway.

**Figure supplement 7.** Heatmap of genes shown in *Figure 2—figure supplement 6* Kyoto Encyclopedia of Genes and Genomes analysis from the DNA replication pathway.

**Figure supplement 8.** Kyoto Encyclopedia of Genes and Genomes (KEGG) for the mismatch repair pathway.

*Figure 2 continued on next page*

*Figure 2 continued*

**Figure supplement 9.** Heatmap of genes shown in *Figure 2—figure supplement 8* Kyoto Encyclopedia of Genes and Genomes analysis from the mismatch repair pathway.

**Figure supplement 10.** Kyoto Encyclopedia of Genes and Genomes (KEGG) for the nucleotide excision repair pathway.

**Figure supplement 11.** Heatmap of genes shown in *Figure 2—figure supplement 10* Kyoto Encyclopedia of Genes and Genomes analysis from the nucleotide excision repair pathway.

analysis defined by direct binding of predictive binding motifs revealed E2F transcription factors as having the highest normalized enrichment score (*Figure 2C*). Because the transcription factor ATF4 was shown to be important for Noxa induction in *Figure 1I*, we compared a known ATF4 gene set (Table S3 from *Wang et al., 2015*), along with an E2F gene set (Table S1 from *Ren et al., 2002*), together with the known p53 gene set and the DEGs in our analog #4 treatment (*Figure 2D*). The resulting Venn diagram of this comparison shows that both ATF4 and E2F targets genes are not unique to these transcription factors and also share common targets with p53 (~5%). Analyzing the ratio of DEGs to the transcription factor gene set did not show an obvious gene enrichment regulation of one transcription factor (*Table 2*). Despite p53 not being the top predictive transcription

**Table 1.** Enriched Biological Process Gene ontology (GO) terms in the 102 differentially expressed genes (DEGs) in CB002-analog #4 treated cells.

GO analysis for the 102 DEGs that are also known p53 target genes. GO term analysis was done using the R package 'goseq' and those genes enriched in particular biological process are described along with their adjp value. Top 25 enriched GO terms are listed.

| GO term ID | Name | Adjp |
|---|---|---|
| GO:0008219 | Cell death | 4.447579E−08 |
| GO:0010941 | Regulation of cell death | 4.447579E−08 |
| GO:0012501 | Programmed cell death | 5.409503E−08 |
| GO:0006915 | Apoptotic process | 5.466988E−08 |
| GO:0043067 | Regulation of programmed cell death | 2.902278E−07 |
| GO:0097193 | Intrinsic apoptotic signaling pathway | 4.205507E−07 |
| GO:0097190 | Apoptotic signaling pathway | 4.949351E−07 |
| GO:0042981 | Regulation of apoptotic process | 6.67383E−07 |
| GO:0072331 | Signal transduction by p53 class mediator | 7.248333E−07 |
| GO:0050896 | Response to stimulus | 7.321667E−07 |
| GO:0007154 | Cell communication | 1.104964E−06 |
| GO:0051716 | Cellular response to stimulus | 1.532609E−06 |
| GO:0023052 | Signaling | 2.587082E−06 |
| GO:0007165 | Signal transduction | 2.587082E−06 |
| GO:0009966 | Regulation of signal transduction | 5.993752E−06 |
| GO:0072332 | Intrinsic apoptotic signaling pathway by p53 class | 1.671799E−05 |
| GO:0048583 | Regulation of response to stimulus | 3.853858E−05 |
| GO:2001233 | Regulation of apoptotic signaling pathway | 3.893307E−05 |
| GO:0010646 | Regulation of cell communication | 4.168569E−05 |
| GO:0007166 | Cell surface receptor signaling pathway | 4.430113E−05 |
| GO:0023051 | Regulation of signaling | 5.136280E−05 |
| GO:0010942 | Positive regulation of cell death | 8.303541E−05 |
| GO:0043065 | Positive regulation of apoptotic process | 1.220935E−04 |
| GO:0048584 | Positive regulation of response to stimulus | 1.220935E−04 |
| GO:0009968 | Negative regulation of signal transduction | 1.220935E−04 |

**Table 2.** Contribution of transcription factors P53, ATF4, and E2F to differentially expressed genes (DEGs) in CB002-analog #4 treated cells.

The total number of DEGs that overlapped with known genes of each transcription factor was calculated. This total is reflected in the 'number of genes in DEG' column. Using this number, we then calculated the ratio of DEGs divided by the total of genes in the transcription factor data set.

| Transcription factor | Number of genes in DEG | Number in data set | Ratio |
|---|---|---|---|
| P53 | 73+10+2+17=**102** | 343 | 0.3 |
| ATF4 | 127+10+19+2=**158** | 559 | 0.28 |
| E2F | 17+2+19+210=**248** | 1444 | 0.17 |

factor in our analysis, ingenuity pathway analysis (IPA) determined p53 to be activated as an upstream regulator with a z-score value of 3.3 and p-value of $2.9 \times 10^{-34}$. A Kyoto Encyclopedia of Genes and Genomes (KEGG) analysis for the p53-pathway signaling was obtained with an adjusted p-value (adjp) equal to $1.18 \times 10^{-1}$ that despite not reflecting a significant enrichment of the p53 pathway, it indicates the presence of a total of 31 DEGs out of the 52 genes tested and present in the KEGG analysis. Thus, this accounts for 60% of DEGs in the KEGG p53-pathway analysis. DEGs involved in the KEGG analysis fold change is described by color and additional genes not shown in the p53-pathway figure and yet involved in the KEGG analysis are shown as a heatmap (*Figure 2— figure supplement 2* and *Figure 2—figure supplement 3*). In line with the GO terms results, p53 target genes involved in apoptosis such as Noxa, Puma, and DR5 were upregulated by the analog #4 treatment. Taken together, this data indicates that although a large set of genes differentially expressed are not predicted to be directly regulated through direct p53 binding, a subset of these are enriched in the p53-pathway, indicative of p53-pathway restoration.

We determined the enriched pathways in the whole set of DEGs (3362). To this end, a KEGG analysis was performed. The top four enriched pathways that were obtained from the KEGG analysis namely included cell cycle, DNA repair, mismatch repair, and nucleotide excision repair. The adjp for each KEGG pathway was $2.27 \times 10^{-6}$, $2.27 \times 10^{-6}$, $5.05 \times 10^{-3}$, and $2.18 \times 10^{-2}$, respectively. The adjp values indicate a significant enrichment score of each pathway. The fold change of DEGs by analog #4 treatment in the KEGG analysis is reflected by the color legend (*Figure 2—figure supplement 4*, *Figure 2—figure supplement 6*, *Figure 2—figure supplement 8*, and *Figure 2—figure supplement 10*). Additional genes not shown in the pathway KEGG figures and yet involved in the KEGG analysis are shown as a heatmap (*Figure 2—figure supplement 5*, *Figure 2—figure supplement 7*, *Figure 2—figure supplement 9*, and *Figure 2—figure supplement 11*). GO terms in biological processes also reflected enrichment of genes that participate in cell cycle regulation (*Table 3*). Taken together, KEGG analysis and GO ontology both reflected the downregulation of genes involved in the G1/S-phase of the cell cycle in CB002-analog treated cells. E2F is responsible for the induction of genes in DNA initiation and replication, such as minichromosome maintenance (MCM) complexes and origin replication complexes (*Bracken et al., 2004*). The transcriptomic analysis indicates the downregulation of these genes and this suggests the inhibition of E2F transcriptional activity. In addition, downregulation of Cyclin E and Cyclin A genes further confirmed the delay of cells to S-phase. GADD45, a p53-target gene that can induce cell cycle arrest, was upregulated. Further study is necessary in order to validate the direct implication of E2F's and p53 target genes in the perturbation of the delay in S-phase. Nonetheless, this data suggests that the identified family of small molecules represent a unique mechanism of action that involves S-phase delay perturbation and p53-pathway restoration.

In order to show that the stimulation of the p53 pathway at the transcriptional level was restoring the p53 pathway at the protein level, a comparative label-free quantitative proteomic analysis of SW480 colon cancer cells in response to DMSO and analog #4 (T4) treated for 24 hr was performed. *Figure 3—figure supplement 1A and E* shows close clustering of protein abundance of each replicate under the same group and variability among the treatments. Volcano plots of fold change versus q-value of the total of 3743 proteins quantified from SW480 cells in response to DMSO, CB002 (CB), and analog #4 (T4) treatments show differentially expressed proteins determined as significant (p<0.05) up and down (*Figure 3—figure supplement 1B–D*). At the protein level, pathway analysis

**Table 3.** Enriched biological process Gene ontology (GO) terms in the 3362 differentially expressed genes (DEGs).

GO analysis for all DEGs by analog #4 treatment. GO term analysis was done using the R package 'goseq' and those genes enriched in particular biological process are described along with their adjp value. Top 20 enriched GO terms are listed.

| GO term ID | Name | Adjp |
| --- | --- | --- |
| GO:0022402 | Cell cycle process | 7.751840E−16 |
| GO:0000278 | Mitotic cell cycle | 7.751840E−16 |
| GO:0007049 | Cell cycle | 2.753525E−15 |
| GO:1903047 | Mitotic cell cycle process | 2.753525E−15 |
| GO:0044770 | Cell cycle phase transition | 3.654863E−13 |
| GO:0006260 | DNA replication | 5.207792E−13 |
| GO:0044772 | Mitotic cell cycle phase transition | 1.192425E−11 |
| GO:0006261 | DNA-dependent DNA replication | 2.919748E−11 |
| GO:0007059 | Chromosome segregation | 3.095763E−11 |
| GO:0044786 | Cell cycle DNA replication | 3.986797E−11 |
| GO:0051301 | Cell division | 1.520253E−10 |
| GO:0000280 | Nuclear division | 1.693587E−09 |
| GO:0098813 | Nuclear chromosome segregation | 1.985723E−09 |
| GO:0033260 | Nuclear DNA replication | 2.161431E−09 |
| GO:0000819 | Sister chromatid segregation | 1.299694E−08 |
| GO:0044843 | Cell cycle G1/S-phase transition | 5.548849E−08 |
| GO:0071103 | DNA conformation change | 6.704071E−08 |
| GO:0048285 | Organelle fission | 7.309501E−08 |
| GO:0051726 | Regulation of cell cycle | 9.496568E−08 |
| GO:0000070 | Mitotic sister chromatid segregation | 1.696716E−07 |

did not reflect an enrichment in p53 targets (*Figure 3A*). Consistent with the microarray data, the proteomic pathway analysis of the differentially abundant proteins shows the downregulation of proteins involved in cell cycle regulation (*Figure 3B*). In particular, CDK4, CKS1B, ERCC6L, MAPK3, and MAX are significantly decreased in analog #4 treatment than in CB002 (*Figure 3C*).

As the CB002-analog molecules were discovered as p53 pathway restoring compounds, we compared the proteomic data, with the known p53 target gene set used in our transcriptomic analysis (Table S3 from Fisher's analysis in *Fischer, 2017*) together with our in-house p53-proteomic database (*Tian et al., 2020*). Our in-house proteomic database was derived from a comparison of HCT116 versus HCT116 p53$^{-/-}$ cells treated with 5-Fluorouracil (5-FU). Our results show that out of all significantly upregulated expressed proteins, only four overlapped with the known p53 targets and six proteins with our in-house p53-proteomics (*Figure 4A*). Eleven proteins were found to be downregulated by analog #4 treatment overlapping with the in-house proteomic database and none with the known p53 target data set (*Figure 4B*). No upregulated or downregulated proteins were found to overlap in all three data sets: analog #4 treatment and both reference databases (*Figure 4A–B*). Overall, these results suggest that within the proteins tested in the proteomic analysis, those expressed by analog #4 treatment and involved in the p53 pathway were minimal under the performed experimental conditions. Additional proteins validated by Western blots, such as Noxa and DR5, were not detected in the proteomic analysis indicating that the proteomic analysis should be considered as preliminary and warrants further optimization. Moreover, differences were observed at the level of protein expression between parental compound CB002 and its analog #4 both downregulated and to a lesser extent, upregulated proteins (*Figure 4—figure supplement 1*). This indicates that these small molecules can have different effects in tumor cells, albeit they have >50% homology in their proteomic composition.

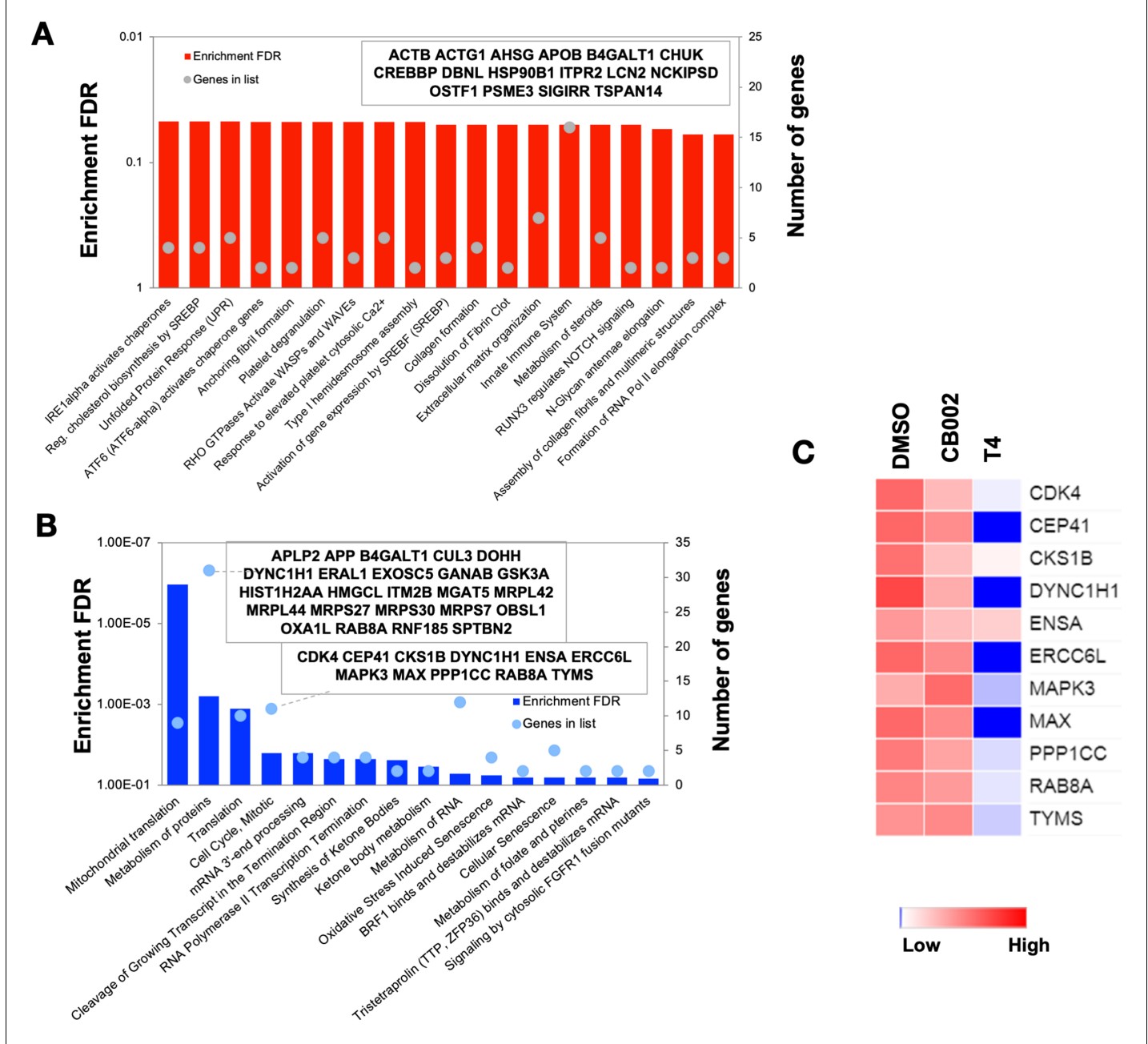

**Figure 3.** Proteomic pathway analysis of CB002-analog #4 responsive differentially expressed proteins in SW480 cells. Significantly enriched pathways corresponding to the CB002-analog #4 responsive upregulated (**A**) and downregulated (**B**) proteins (in comparison with the DMSO). The heatmap (**C**) shows the grouped proteins' expression value of some target pathway proteins highlighted in the box area. Data collected from the proteomic analysis of DMSO versus CB002 and analog #4 treated SW480 cell samples for 24 hr.

The online version of this article includes the following source data and figure supplement(s) for figure 3:

**Source data 1.** Protein information of all proteins detected in DMSO vehicle control and analog #4 SW480 treated cells at 24 hr samples analysed by LC-MS/MS.

**Figure supplement 1.** Comparative label-free quantitative proteomic analysis of SW480 cell lines in response to DMSO, CB002 (CB), and analog 4 (T4) treated for 24 hr.

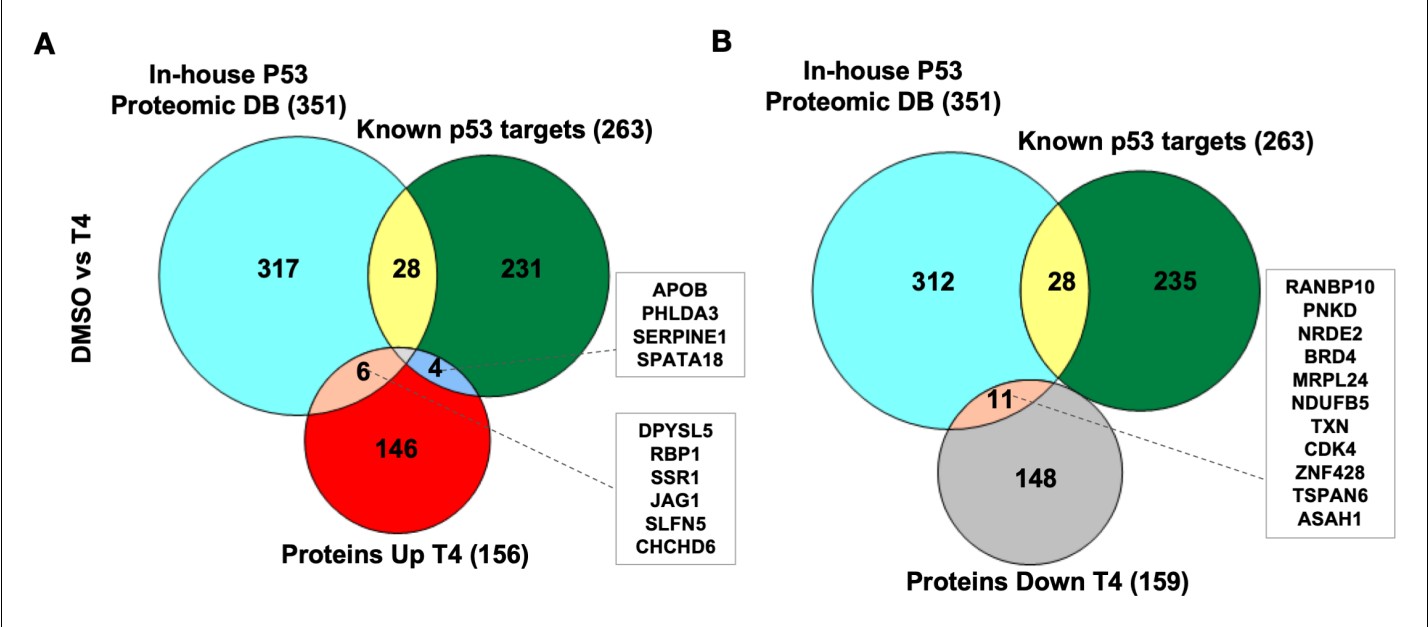

**Figure 4.** CB002-analog #4 (T4) responsive proteins in comparison with in-house p53-proteomic database and known p53 targets. Three-way Venn diagram of upregulated (A) and downregulated (B) analog #4 responsive proteins. Data collected from the proteomic analysis of DMSO versus analog #4 treated SW480 cell samples for 24 hr.

The online version of this article includes the following figure supplement(s) for figure 4:

**Figure supplement 1.** Proteomic data comparison of proteins increased and/or decreased in abundance with analog #4 (T4) treatment compared to DMSO and CB002.

## CB002 and analogs perturb an S-Phase but not G2 checkpoint, unlike other xanthines

Caffeine is a G2 checkpoint deregulator through inhibition of ATM/ATR. Thus, the combination of chemotherapy agents with caffeine results in enhanced cancer cell cytotoxicity. Nonetheless, it was not pursued due to caffeine's lack of achievable required concentrations in plasma. We investigated whether CB002 and its analogs can deregulate the G2 checkpoint, like caffeine, pentoxifylline, and theophylline. We synchronized SW480 colon cancer cells using double thymidine block, released and treated with CB002-analog compound alone or in combination with etoposide, and probed for key G2/M-phase cell cycle markers. As expected, we observed that etoposide treatment enhances protein expression of pcdc2$^{(Tyr15)}$ and pcdc25c$^{(Ser16)}$ indicating cell cycle arrest due to DNA damage. The combination of etoposide with caffeine resulted in G2-deregulation as indicated by decreased expression of pcdc2$^{(Tyr15)}$ and pcdc25c$^{(Ser16)}$. Similarly, the combination of etoposide with CB002 or CB002-analog #4 showed a decrease in expression of pcdc2$^{(Tyr15)}$ and pcdc25c$^{(Ser16)}$. Nonetheless, CB002 or CB002-analog #4 do not increase M-phase marker pH3$^{(Ser10)}$ as would be expected for a G2-deregulator like caffeine (*Figure 5A*). This data suggests that CB002 and CB002-analog #4 either do not deregulate the G2 checkpoint or that these compounds delay cells going into M-phase. Moreover, CB002 and its analogs increase p-Cdc25c and p-Cdc2 in combination with etoposide indicating cell cycle arrest. A similar experiment was performed as a time course after cell synchronization release to further elucidate the cell cycle effects of CB002-analog #4. As seen in *Figure 5D*, cell cycle markers pcdc2$^{(Tyr15)}$ and pcdc25c$^{(Ser16)}$ expression decreased in CB002-analog #4 compared to DMSO and etoposide and their expression over time increased at 12 hr indicative of a delay of cells in the G2 cell cycle phase. To further elucidate the effect in S-phase, we evaluated Cyclin A and p-RPA-RPA2$^{(S8)}$, the latter as a marker of single-stranded DNA and replication stress that are potentially caused by stalled or collapsed replication forks. Cyclin A expression did not decrease over time in CB002-analog #4 treated cells as compared to DMSO and etoposide indicating that cells were delayed in S-phase. Moreover, p-RPA-RPA2$^{(S8)}$ expression upon CB002-analog #4 treatment was increased compared to DMSO indicating replication stress. The p53 target p21 was also found to

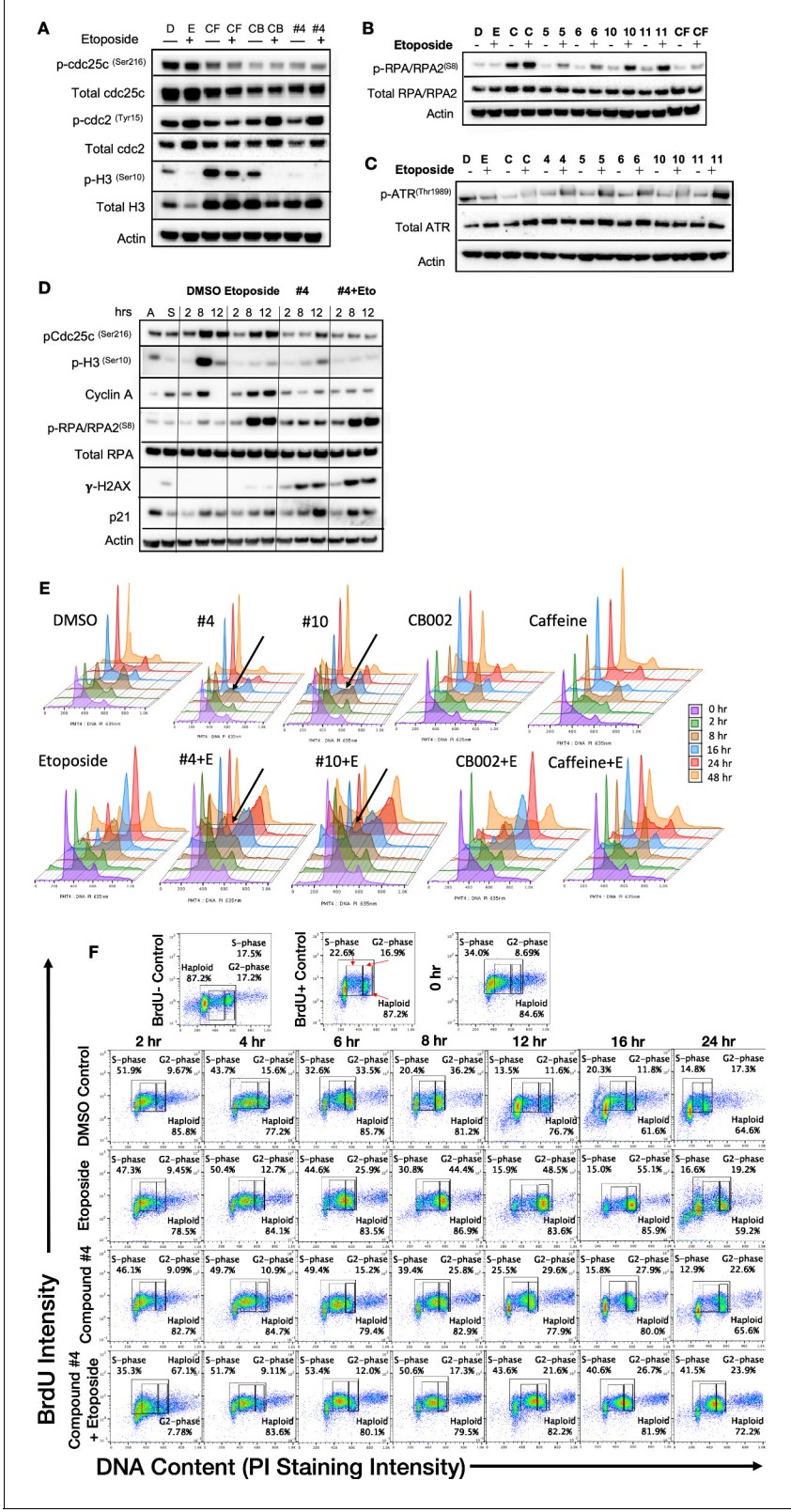

**Figure 5.** CB002 and its analogs perturbed an S-phase rather than a G2-phase checkpoint like other known xanthines cell cycle effects in SW480 cells. Western blot analysis of synchronized SW480 treated cells as indicated and harvested at 24 hr (**A, B, C**). Synchronized SW480 cells were treated as indicated and analyzed by Western blot (**D**), PI staining (**E**) or PI/BrdU analysis (**F**). CB002 (C), Caffeine (CF). Experiments from figures (**A**)–(**D**) were performed at least twice and a representation of one is shown.

*Figure 5 continued on next page*

*Figure 5 continued*

The online version of this article includes the following figure supplement(s) for figure 5:

**Figure supplement 1.** Flow cytometry PI/BrdU-CB002-analog #4 perturbs the S-phase rather than the G2 checkpoint, unlike other xanthines.

increase in CB002-analog #4 treated cells indicating cell cycle arrest. Taken together, these analogs deregulate an S-phase checkpoint and not a G2 checkpoint.

To investigate further the effects of these CB002-analogs on the cell cycle, we probed for S-phase specific markers and performed PI analysis by flow cytometry upon release of synchronized cells for a time course of 0–48 hr. CB002 and its structural analogs, unlike caffeine, increase single-strand DNA marker p-RPA-RPA2[S8] and p-ATR[Thr1989], indicating that these compounds result in replication stress and activate features of an S-phase checkpoint (*Figure 5B–C*). PI analysis further confirms that combination of caffeine and etoposide deregulates the G2 checkpoint and that CB002-analogs #4 and #10 treatment results in S-phase accumulation are particularly observed at 8 hr following release from synchronization (*Figure 5E*). PI and BrdU co-staining confirm that CB002-analog #4 increases by 30% cells in S-phase at 12 hr as compared to DMSO vehicle control and no significant differences are observed in G2-phase cells between etoposide and CB002-analog #4 at 24 hr (*Figure 5F*). S-phase delays with CB002 and CB002-analog #10 occur at 6–8 hr of treatment, particularly a two-fold difference in combination with etoposide. The caffeine-treated S-phase population is comparable to the DMSO vehicle control at all time points indicating that caffeine does not perturb the S-phase. As expected, caffeine decreases the G2-population by 2- to 3-fold at 24 hr in combination with etoposide as compared to etoposide alone, and no other treatment tested decreases the G2-population when combined with etoposide (*Figure 5* and *Figure 5—figure supplement 1*). Haploid cell gating indicates the haploid BrdU-positive cells in *Figure 5* and *Figure 5—figure supplement 1*.

## CB002-analog #4 has anti-tumor effects in vitro and in vivo

We focused on lead CB002-analog #4 and investigated its therapeutic index in vitro and in vivo. We treated an isogenic HCT116 cell line panel with varying p53 mutation-status were treated with 100 µM CB002 and 25 µM CB002-analog #4 and established IC50 values by the Cell-Titer glow cytotoxicity assay. Across this panel, CB002-analog #4 has a 20- to 30-fold range in IC50 values, independently of the HCT116 p53-status (*Figure 6A*). Thus, the results indicate that the restoration of the p53-pathway by CB002 or analog #4 is p53-independent. SW480 cells treated with CB002-analog #4 showed a significant increase of sub-G1 content as compared to vehicle control, whereas treatment with CB002-analog #4 of normal human WI38 lung fibroblast cells did not significantly increase the sub-G1 cell population indicating that CB002-analog #4 is safe to normal cells in vitro (*Figure 6B*).

We further investigated the anti-cancer cytotoxicity potential of CB002-analog #4. We treated a colorectal cancer patient-derived organoid with CB002-analog #4 and performed cellular cytotoxicity analysis in vitro and immunofluorescence staining of ethidium homodimer, calcein, caspase-3, and Ki-67 to distinguish between dead, live, apoptotic, and proliferating cells, respectively. CB002-analog #4 enhances cytotoxicity as compared to the CB002 parent compound in the tested colorectal cancer patient-derived organoid as indicated by the cell viability response curve (*Figure 6C*). Moreover, the immunofluorescence assay staining for ethidium homodimer and calcein shows an increase of ethidium homodimer staining of CB002 and CB002-analog #4 to a larger extent as compared to vehicle control indicating an enhanced killing of cells. Calcein staining shows that organoids treated with CB002-analog #4 are smaller in size indicating that CB002-analog #4 decreases the growth of the patient-derived organoid (*Figure 6D*). Cleaved caspase-3 staining indicates that both CB002 and CB002-analog #4 treatment at IC50 doses increases apoptotic cells (*Figure 6D*). CB002-analog #4 treatment also results in an inverse relationship with Ki-67 staining with respect to drug concentration, indicating that CB002-analog #4 decreases the population of proliferating cells (*Figure 6E*).

We investigated CB002-analog #4 in vivo for anti-tumor efficacy as well as toxicity in NSG mice. Mice were xenografted with human SW480 colorectal cancer cells treated with CB002-analog #4 at 50 mg/kg by oral gavage three times per week. Our data suggests that CB002-analog #4 is well tolerated as indicated by the mouse body weights during the duration of the experiment (*Figure 6F*). At 5 weeks of treatment, CB002-analog #4 treated tumors have a statistically significant

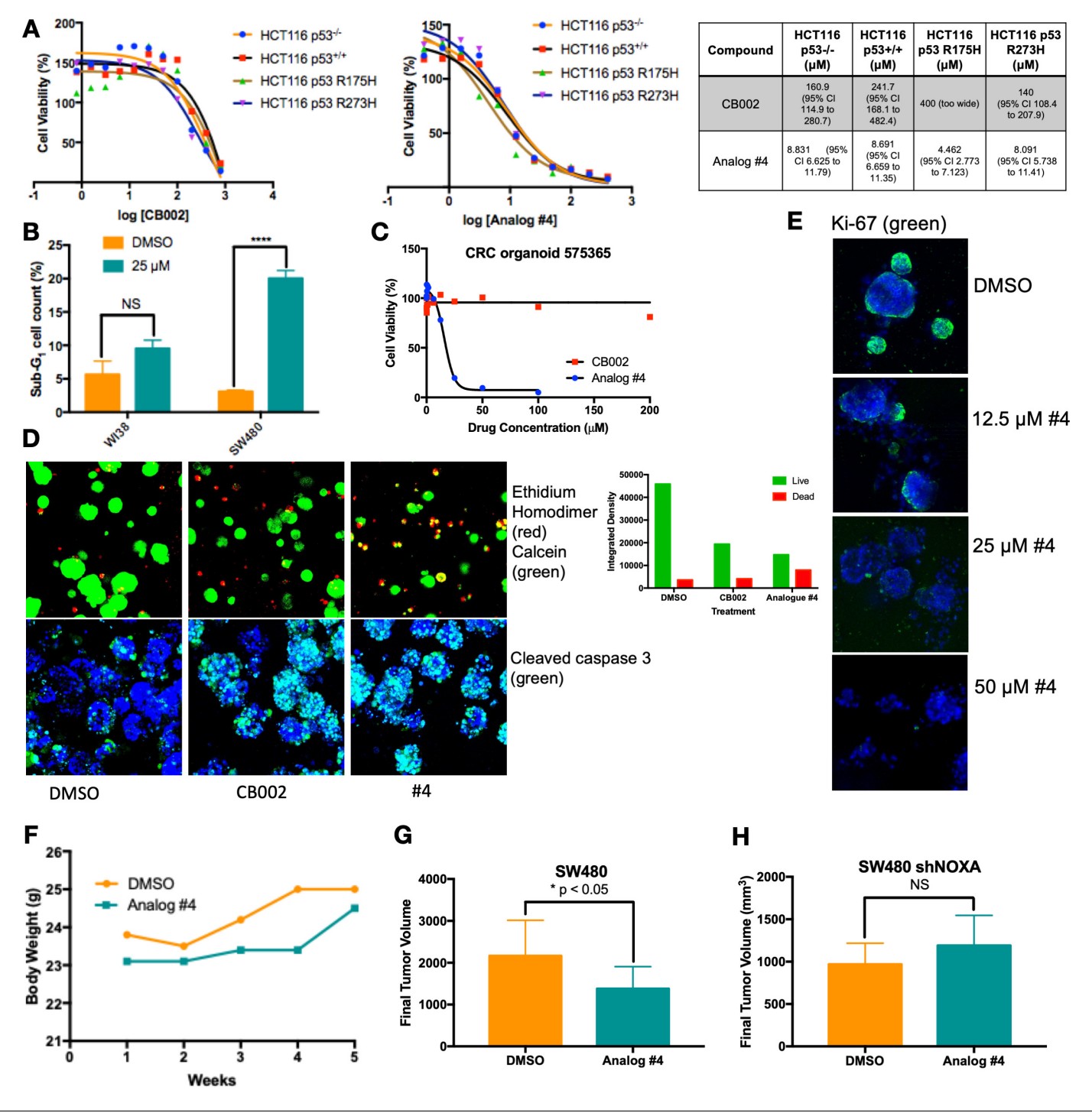

**Figure 6.** CB002-analog #4 has potent anti-tumor effects in vitro and in vivo. HCT116 isogenic panel treated with CB002 or analog #4 for 48 hr and their respective IC50 values shown in the table (**A**). CB002-analog #4 increases apoptotic cells as indicated by the sub-G1 content in cancer cells but not in normal WI38 cells (48 hr). Two-way ANOVA, p<0.0001 (**B**). 72 hr treatment with CB002-analog #4 is most potent (**C**) and increases dead cells as indicated by the ethidium homodimer staining (red) compared to calcein stained live cells (green) (**A**), and cleaved caspase-3 (green) immunofluorescence (**D**) in colorectal cancer patient-derived organoid cells. CB002-analog #4 decreases ki67 staining (green) in a dose-dependent manner (72 hr) in colorectal cancer patient-derived organoid cells (**E**). CB002-analog #4 is non-toxic in vivo (**F**) and significantly reduces tumor volume in NSG mouse xenografts with SW480 wild-type cells (**G**) but not in SW480 cells with shNoxa (**H**). 50 mg/kg by oral gavage three times per week, final tumor volume at 5 weeks. Unpaired t-test, p<0.05.

lower tumor volume as compared to vehicle control (*Figure 6G*). To determine the importance of Noxa in vivo, mice were xenografted with SW480 cells containing a stable knockdown of Noxa. Mice xenografted with SW480 shNoxa cells did not have a significant difference in tumor volume after CB002-analog #4 treatment compared to vehicle control treated tumors indicating that Noxa is important for reduced tumor volume in vivo (*Figure 6H*).

## Discussion

We describe a novel class of anti-tumor agents with a unique mechanism of action involving restoration of p53 pathway signaling, independently of p53, in tumors with mutated-p53 and characteristics of an S-phase checkpoint. The defining members of this class that best exemplify the novel mechanistic properties are CB002-analogs #4 and #10. The properties of these CB002-analog xanthine compounds are different from other xanthines, such as caffeine, pentoxifylline, and theophylline, that do not restore p53 pathway signaling in tumors with mutant p53 and which deregulate a G2 checkpoint rather than induce an S-phase checkpoint.

Our approach to discovering p53 pathway restoring compounds involved cell-based screening for functional restoration of p53-regulated reporter activity, coupled with cell death induction. Thus, small molecule lead compounds and structural-analogs were not expected to act directly on mutant p53 or restore binding of mutant p53 to genes normally regulated by p53. In the case of the compounds described here, activation of p53 target genes such as Noxa or DR5 occurred independently of p53 and this was observed in tumor cells with different p53 mutations. Thus, there is no expectation that CB002 or analogs #4 or #10 will cause mutant p53 to bind to DNA or chromatin in the regulatory regions of Noxa or DR5 in a manner that wild-type p53 does. Moreover, the induction of p53 targets occurred independently of p53 family member p73, but in a manner that requires integrated stress response transcription factor proteins ATF3/4. These results provide a molecular mechanism for activation of p53 target genes in a manner that substitutes transcription factors such as ATF3/4 for defective p53. This mechanism results in tumor suppression through induction of pro-apoptotic factors despite p53 mutation, and therefore acts as a bypass mechanism to prevent tumor growth in drug-treated cells.

CB002-analog #4 is 20–30 times more potent and like the CB002 parental compound restores the p53-pathway and induces apoptosis independently of p73. The 12 p53 pathway restoring structural analogs of CB002 tested were similar in that they resemble the structure of a xanthine. Our transcriptional analysis identified 102 genes involved in the p53-pathway and IPA determined p53 to be activated as an upstream regulator with a z-score value of 3.3 and p-value of $2.9 \times 10^{-34}$. This data further validates the novel anti-cancer class of small molecules as p53-restoring drugs. Microarray analysis identified approximately 150 genes involved in cell cycle regulation, DNA synthesis, and repair that are significantly decreased compared to DMSO control. These genes include, minichromosome maintenance (MCM) proteins, Cyclin E, CDK, E2F, and Cdc2 (*Figure 2—figure supplements 4–11*). Proteomic analysis also confirmed a decrease in proteins involved in cell cycle regulation (*Figure 3B*). Thus, our transcriptomic and proteomic analyses coincide in that CB002-analog #4 significantly reduces key regulators of the cell cycle. Taken together with the fact that known xanthines such as caffeine deregulate the G2 checkpoint, we examined the effects of the CB002-analogs on the cell cycle. Our data indicate that the p53-restoring CB002-analog compounds, unlike known xanthines such as caffeine, pentoxifylline, and theophylline, restore the p53 and do not deregulate the G2 checkpoint. Instead, treatment with these small molecule CB002-analogs results in activation of an S-phase DNA damage response pathway characterized by the increase in p-ATR$^{(Thr1989)}$ and we suggest this ultimately leads to a delay of cells in S-phase and this S-phase perturbation may contribute to cancer cell death. Importantly, the observed S-phase perturbation may lead to new therapeutic regimens such as synthetic lethality in BRCA-deficient cells and combination with PARP inhibitors.

We previously reported that pro-apoptotic protein Noxa plays a critical role in CB002-mediated cell death. Our data shows that CB002-analogs induce Noxa expression across different colorectal cancer cell lines in vitro. More importantly, we show that Noxa appears to be critical in vivo as CB002-analog #4 treatment of SW480 shNoxa tumors does not significantly reduce tumor volume as compared to vehicle control. We have evidence indicating that ATF3/4 play a role in regulating Noxa as knockdown of ATF3/4 results in the decrease of Noxa protein expression. Our proteomic

data shows activation of the integrated stress response as indicated by the increase of genes involved in the unfolded protein response, tRNA aminoacylation, and increase of ATF3/4 protein expression by Western blot (*Figure 3A*, *Figure 1I*). Whether the S-phase perturbation is a result of cellular stress remains to be addressed.

ATF3/4 can regulate similar targets of that of p53, including p21. Our laboratory has identified a small molecule PG3-Oc which involves the restoration of the p53 pathway independently of p53 through ATF4 (*Tian et al., 2021*). P53 has been shown to indirectly repress many cell cycle genes through the induction of p21. P21, in turn, binds to the DREAM repressor complex which represses genes controlled by E2Fs and CHR transcription factors (*Fischer et al., 2016*; *Engeland, 2018*). We observed many cell cycle genes downregulated at the transcriptional level that are relevant to the p53 signal pathway. Moreover, our bioinformatic analysis predicted E2Fs as one of the transcription factors. We have previously shown that CB002 induces p21 expression (*Hernandez-Borrero et al., 2018*), as well as analog #4 in this study thus it is possible that the observed S-phase perturbation is through p53-independent p21 stimulation that binds to DREAM complexes. Therefore, it will be interesting to see if ATF3/4 regulate p21 expression and the effect of p21 knockdown on cell cycle genes and affect the S-phase perturbation observed by CB002-analogs.

We show that CB002-analog #4 induces apoptosis in colorectal cancer patient-derived organoid cells and that it is safe both in vitro and in vivo as indicated by the lack of a statistically significant increase in the sub-G1 population in normal human fibroblasts and also a healthy NSG mice body weight throughout treatment, respectively. The observed decrease in tumor volume was statistically significant at 5 weeks. This effect was suboptimal than desired and further optimization will be required to reach optimal effects. Importantly, the decrease in tumor volume by CB002-analog #4 is dependent on Noxa. As Noxa is not commonly mutated in human cancer, its induction by the CB002-analogs offers a feasible therapeutic advantage leading to tumor cell death and its expression may be used as a pharmacodynamic biomarker to predict therapeutic response. Taken together, our data suggests that CB002-analogs #4 and #10 represent a novel class of anti-tumor agents that provide a unique therapeutic strategy that can be clinically translated.

## Materials and methods

### CB002-analog small molecule secondary drug screening

CB002 structural analogs were obtained from ChemBridge Library and screening was performed in the human SW480 colorectal cancer cell line that stably expresses a p53-regulated luciferase reporter previously generated in our laboratory (*Wang et al., 2006*). Cells were seeded at a density of $1 \times 10^4$ cells per well in 96-well plates (Greiner Bio-One) and treated with the indicated compound from 0 to 600 μM. p53 transcriptional activity was imaged using an IVIS imaging system at 6 hr. A total of three biological replicates per condition were performed.

### Cell lines and culture conditions

DLD-1 (p53$^{S241F}$) (RRID:CVCL_0248), SW480 (p53$^{R273H,P309S}$) (RRID:CVCL_0546), and HCT116 (p53$^{WT}$) (RRID:CVCL_0291) colorectal cancer cell lines and WI38 normal lung fibroblast cells were purchased from ATCC. HCT116 p53$^{-/-}$ (obtained from the Vogelstein Laboratory, Johns Hopkins University), HCT116 R175H p53, and HCT116 R273H p53 were previously described (*Hernandez-Borrero et al., 2018*). The SW480 cancer cell line that stably expresses a p53-regulated luciferase reporter was previously generated in our laboratory (*Ren et al., 2002*). Cell lines were authenticated and tested for mycoplasma. Cell lines were maintained in HyClone Dulbecco's High Glucose Modified Eagles Medium (DMEM, GE Healthcare), HyClone McCoy's 5A (GE Healthcare) or Eagle's Minimum Essential Medium (EMEM, ATCC) containing 10% fetal bovine serum (FBS) and 1% penicillin/streptomycin (complete media) at 37°C in 5% $CO_2$, as recommended by ATCC.

### CellTiter-Glo luminescent cell viability assay

SW480 cells were seeded in 96-well plates at a density of $5 \times 10^3$ cells per well. A total of three biological replicates per condition were performed. About 20 μL of CellTiter-Glo reagent was added directly to the wells, according to the manufacturer's protocol, and bioluminescence signal was determined using an IVIS imaging system at a period of 48–72 hr after treatment.

## Cell synchronization

Where indicated, cells were synchronized by double thymidine block. Cells were treated with 2 μM Thymidine for 16 hr, drug was removed and replaced by complete growth media for 8 hr. Cells were treated for the second time with 2 μM Thymidine for 16 hr, at this point, cells were treated and harvested as indicated.

## Propidium Iodide and BrdU flow cytometry assay

Cells were seeded at a density of $5 \times 10^5$ in a six-well plate and treated for 48–72 hr. A total of two biological replicates per condition were performed. After treatment, floating cells were collected and adherent cells were trypsinized, pelleted, washed with phosphate-buffered saline (PBS) and fixed in 70% ethanol overnight. For PI based sub-G1 apoptosis analysis, cells were spun down after fixation and resuspended in phosphate-citric acid buffer (0.2 M $Na_2HPO_4$+0.1 M citric acid, pH 7.8) at room temperature for 5 min. The cell pellet was resuspended for staining with 50 μg/mL PI and 250 μg/mL ribonuclease (RNase A). For BrdU Chase analysis, a final concentration of 10 μM BrdU (Sigma-Aldrich, B9285) was added to the cell culture for 30 min at 37°C prior to cell fixation. Cells were fixed, spun down, and resuspended in 1 mL of 2 N HCL with 0.5% Triton X-100 for 30 min at room temperature. Cells were pelleted, washed with PBS, and resuspended in 20 μL BrdU anti-body (BD Biosciences, cat no. 347580) diluted in 0.5% Tween 20/PBS/5% BSA for 30 min at room temperature. Cells were then spun down and resuspended in 140 μg/mL goat anti-mouse Alexa Fluor 488 (#A-11008, Thermo Fisher Scientific) in 0.5% Tween 20 in PBS/5% BSA for 30 min at room temperature. Cells were then spun down and resuspended in 5 μg/mL PI: 250 μg/mL RNase A solution. Samples were analyzed on an Epics Elite flow cytometer (Beckman Coulter).

For BrdU analysis gating, cell aggregates were gated out in the PI Peak versus DNA PI histogram. BrdU lower limit intensity was set on upper limit of the negative control. No BrdU antibody in *Figure 3E* and no goat anti-mouse Alexa Fluor 488 antibody in *Figure 5—figure supplement 1* were used as the negative controls. Haploid cell gating indicates the haploid BrdU-positive cells. S-phase and G2-phase boundaries were determined by PI staining that indicated G1 and G2 as per DNA content. Gating was held constant throughout the samples within a given experiment.

## Immunoblotting

After treatment, floating cells were collected and adherent cells were trypsinized, washed with PBS, and lysed with RIPA buffer (Sigma-Aldrich) for 30 min to 1 hr at 4°C. Protein lysates were pelleted and supernatant was collected. Total protein per sample was determined using a Pierce BCA Protein Assay Kit (Thermo Fisher Scientific). Proteins were denatured using 1× NuPAGE LDS sample buffer (Thermo Fisher Scientific) and reduced with 2-Mercaptoethanol (Sigma-Aldrich). Protein lysates were boiled for 15 min at 95°C. After protein normalization, samples were loaded into NuPAGE Novex 4–12% Bis-Tris Protein Gels (Thermo Fisher Scientific) and gel electrophoresis was performed with NuPAGE MES SDS Running Buffer, with the exception of ATR that was ran using a NuPAGE Novex 3–8% Tris-Acetate Protein Gels (Thermo Fisher Scientific) and NuPAGE Tris-Acetate SDS Running Buffer. Proteins were transferred onto an Immobilon-P membrane (PVDF, EMD Millipore) using a Bio-Rad system with a 10% Tris-Glycine and 10% methanol transfer buffer diluted in distilled and deionized water. Membranes were blocked with 10% milk in TBST solution and then incubated overnight with primary antibody, washed with TBST, and incubated with secondary antibody for 1 hr. Primary antibody incubations were performed in 5% milk or 5% BSA in TBST solution as per manufacturer instructions. Signal was detected using a Chemiluminescent Detection Kit, followed by autoradiography. The following antibodies were used: cytochrome C (1:1000; #sc-13560; Santa Cruz; RRID:AB_627383), p53 (1:1000; #sc-126; Santa Cruz; RRID:AB_628082), p73 (1:1000; #A300-126A; Bethyl Laboratories), Noxa (1:250; #OP180; EMD Millipore; RRID:AB_564933), DR5 (1:1,000; #3696; Cell Signaling Technology; RRID:AB_10692107), cleaved PARP (1:1,000; #9546; Cell Signaling Technology; RRID:AB_2160593), ATF3 (1:1000, #sc-188, Santa Cruz; RRID:AB_2258513), ATF4 (1:1,000; #11815; Cell Signaling Technology), p-RPA32/RPA2$^{(Ser8)}$ (1:1,000; #54762, Cell Signaling Technology), RPA32/RPA2 (1:1000; 52448; Cell Signaling Technology), p-cdc2$^{(Tyr15)}$ (1:1000; #9111; Cell Signaling Technology), cdc2 (1:1000; #54; Santa Cruz Biotechnology), p-cdc25c$^{(Ser216)}$ (1:1000; #9528; Cell Signaling Technology; RRID:AB_2075150), cdc25c (1:1000; #13138; Santa Cruz Biotechnology; RRID:AB_627227), p-H3$^{(Ser10)}$ (1:1000; #3377; Cell Signaling Technology; RRID:AB_1549592),

H3 (1:1000; #14269; Cell Signaling Technology; RRID:AB_2756816), γ-H2AX$^{(Ser139)}$ (1:1000; #2577; Cell Signaling Technology; RRID:AB_2118010), p-ATR$^{(Thr1989)}$ (1:1000; GTX128145, GeneTex; RRID: AB_2687562), ATR (1:1000; #1887; Santa Cruz Biotechnology; RRID:AB_630893), Cyclin A (1:1000; sc-271682, Santa Cruz Biotechnology;), p21 (1:200; #OP64; EMD Millipore; RRID:AB_2335868), Ran (1:10000; #610341; BD Biosciences; RRID:AB_397731), and β-actin (1:10000, A5441, Sigma-Aldrich; AB_476744).

## Knockdown of expression of p73, ATF3, and ATF4 using siRNA

A total of $1\times10^5$ cells/well were plated per well in a 12-well plate in a medium with 10% FBS without antibiotic. Forward transfection of p73 siRNA (s14319, Ambion), ATF3 siRNA (sc-29757), and ATF4 (sc-35112) was performed using the Lipofectamine RNAiMAX Transfection Reagent (Life Technologies) and incubated for 48 hr before drug treatments.

## Microarray analysis

SW480 cells were seeded at a density of $1\times10^6$ in 10 cm dishes and once adhered, treated with DMSO vehicle control or CB002-analog #4 for a total of two biological replicates per condition. Floating cells were collected and adherent cells were trypsinized at 12 hr of treatment. Cells were pelleted and RNA was isolated using a Quick-RNA MiniPrep (#R1055, Zymo Research) according to the manufacturer's instructions. RNA quality was tested using an Agilent Bioanalyzer RNA Kit. Once RNA quality was sufficient, RNA was amplified and labeled using the Low RNA Input Linear Amplification Kit (Agilent). Labeled cDNA was hybridized onto Affymetrix Human Gene 2.0 ST array. Significant changes in gene expression were determined as follows: the low expression cutoff of probe signal intensity was set at 50 (unless at least one sample did not meet these criteria for that particular probe). Normalization was performed using the RMA method and Limma eBayes for the statistical method using R studio programming software. Genes with an FDR of <0.05 were determined as significant in DMSO vehicle control versus analog #4.

## Sample preparation for LC-MS/MS analysis

SW480 cells were seeded at a density of $1\times10^6$ in 10 cm dishes and treated with DMSO vehicle control or CB002-analog #4 for 24 hr. A total of three biological replicates per condition were performed. Floating cells were collected and adherent cells were trypsinized. Cells were spun down, wash with PBS, and pelleted cells were flash frozen with liquid $N_2$ and subjected for for LC-MS/MS analysis.

Briefly, cell pellets were lysed with a lysis buffer (8 M urea, 1 mM sodium orthovanadate, 20 mM HEPES, 2.5 mM sodium pyrophosphate, 1 mM β-glycerophosphate, pH 8.0, 20 min, 4°C) followed by sonication at 40% amplification by using a microtip sonicator (QSonica, LLC, Model no. Q55) and cleared by centrifugation (14,000×$g$, 15 min, 15°C). Protein concentration was measured (Pierce BCA Protein Assay, Thermo Fisher Scientific) and a total of 100 μg of protein per sample was subjected for trypsin digestion. Tryptic peptides were desalted using C18 Sep-Pak plus cartridges (Waters, Milford, MA) and were lyophilized for 48 hr to dryness. The dried peptides were reconstituted in buffer A (0.1 M acetic acid) at a concentration of 1 μg/μL and 5 μL was injected for each analysis.

The LC-MS/MS was performed on a fully automated proteomic technology platform that includes an Agilent 1200 Series Quaternary HPLC system (Agilent Technologies, Santa Clara, CA) connected to a Q Exactive Plus mass spectrometer (Thermo Fisher Scientific, Waltham, MA). The LC-MS/MS setup was used as described earlier (*Ahsan et al., 2017*). Briefly, the peptides were separated through a linear reversed-phase 90 min gradient from 0% to 40% buffer B (0.1 M acetic acid in acetonitrile) at a flow rate of 3 μL/min through a 3 μm 20 cm C18 column (OD/ID 360/75 μm, Tip 8 μm, New objectives, Woburn, MA) for a total of 90 min run time. The electrospray voltage of 2.0 kV was applied in a split-flow configuration, and spectra were collected using a top 9 data-dependent method. Survey full-scan MS spectra (m/z 400–1800) were acquired at a resolution of 70,000 with an AGC target value of $3\times106$ ions or a maximum ion injection time of 200 ms. The peptide fragmentation was performed via higher-energy collision dissociation with the energy set at 28 normalized collision energy. The MS/MS spectra were acquired at a resolution of 17,500, with a targeted value of $2\times10^4$ ions or maximum integration time of 200 ms. The ion selection abundance threshold was set

at $8.0 \times 10^2$ with charge state exclusion of unassigned and z=1, or 6–8 ions and dynamic exclusion time of 30 s.

## Database search and label-free quantitative analysis

Peptide spectrum matching of MS/MS spectra of each file was searched against the NCBI Human database (TaxonID: 9606, downloaded on 02/19/2020) using the Sequest algorithm within Proteome Discoverer v 2.4 software (Thermo Fisher Scientific, San Jose, CA). The Sequest database search was performed with the following parameters: trypsin enzyme cleavage specificity, two possible missed cleavages, 10 ppm mass tolerance for precursor ions, and 0.02 Da mass tolerance for fragment ions. Search parameters permitted dynamic modification of methionine oxidation (+15.9949 Da) and static modification of carbamidomethylation (+57.0215 Da) on cysteine. Peptide assignments from the database search were filtered down to a 1% FDR. The relative label-free quantitative and comparative among the samples were performed using the Minora algorithm and the adjoining bioinformatics tools of the Proteome Discoverer 2.4 software. To select proteins that show a statistically significant change in abundance between two groups, a threshold of 1.5-fold change with p-value (0.05) was selected.

## Immunohistochemistry

30,000 cells/well were seeded in eight-chamber slides. Cells were washed with PBS at the harvesting time point and fixed with 4%parafornaldehyde for 25 min. Cells were then washed with PBS and permeabilized with 0.2% Triton X-100 for 5–10 mins. Cells were then washed with PBS and incubated overnight 1:100 with the indicated primary antibody cytochrome C (#sc-13560; Santa Cruz; RRID: AB_627383), Tom-20 (#42406, Cell Signaling Technology; RRID:AB_2687663), cells were washed with PBS and incubated with secondary antibody 1:200 goat anti-mouse Alexa Fluor 488 (#A-11008, Thermo Fisher Scientific; RID:AB_143165) and Cy3 AffiniPure Donkey anti-rabbit (#711-165-152, Jackson Immuno Research) for 1 hr followed by PBS washed, 1:400 DAPI staining, washed with PBS and imaged. Organoid viability imaging was determined by CellTrace Calcein Green (#C34852, Thermo Fisher Scientific), Ethidium Homodimer-1 (#E1169, Thermo Fisher Scientific), Ki-67 (#9449; Cell Signaling Technology; RRID:AB_2797703) incubated at 37°C for 1 hr then washed with PBS and imaged. Imaging was done using a Leica Confocal Microscope. Experiments were performed at least twice and more than three technical replicates were obtained, a representation of one is shown.

## Drug efficacy using in vivo tumor xenografts

In vivo drug efficacy studies were performed on 10 NSG (RRID:IMSR_JAX:005557) female randomized mice per cohort. Mice tested negative for pathogens listed on Indexx Bioanalytics Laboratory IMPACT I testing including *Mycoplasma* spp., *Mycoplasma pulmonis*, mouse hepatitis virus, pneumonia virus, murine norovirus, sendai virus, and *Corynebacterium bovis*. Tumor inoculation was induced by subcutaneous injection in the left and right dorsal flank, each with a 150 μL suspension of 1–$5 \times 10^6$ human colon cancer cells in PBS with Matrigel (1:1). Once tumor size reached 100 mm³, mice were treated 3×/week with DMSO vehicle or compound #4 via oral gavage (22 gauge 1 in. needle) in a solution of 10% DMSO, 20% Kolliphor EL (Sigma-Aldrich, cat. no. C5135) and 70% PBS. Mouse weight and tumor measurements were recorded 1–2 times per week. Tumor volume was calculated as V=0.5*L*W2, were L is length and W is width of the tumor. At the end of the experiment, mice were euthanized by $CO_2$. All in vivo procedures were performed according to an approved Institutional Animal Care and Use Committee (IACUC) protocol #14–17 at Fox Chase Cancer Center.

## Statistical analysis

To assess the statistical significance, two-way ANOVA or unpaired t-test for two comparisons was performed, with $p < 0.05$ defined as statistically significant. Data are presented as means ± SEM (three biological replicates). Comparisons were made against the DMSO vehicle control.

## Disclosure of potential conflicts of interest

WSE-D is a Founder of p53-Therapeutics, Inc, a biotech company focused on developing small molecule anti-cancer therapies targeting mutant p53. WSE-D has disclosed his relationship with p53-

Therapeutics and potential conflict of interest to his academic institution/employer and is fully compliant with NIH policies and institutional policies that is managing this potential conflict of interest.

## Acknowledgements

This work was presented in part at the annual American Association for Cancer Research (AACR) meetings in 2017, 2018, 2019, and 2020. LJHB received the AACR Minority Scholar Research Award in 2017. This work was supported in part by NIH grants CA176289 (WSE-D). WSE-D is an American Cancer Society (ACS) Research Professor and is supported by the Mencoff Professorship in Medical Science at Brown University. This work was supported by the American Cancer Society and by the Teymour Alireza P'98, P'00 Family Cancer Research Fund established by the Alireza Family.

## Additional information

### Competing interests

Wafik S El-Deiry: Founder and shareholder of p53-Therapeutics, Inc, a biotech company focused on developing small molecule anti-cancer therapies targeting mutant p53. WSED has disclosed his relationship with p53-Therapeutics and potential conflict of interest to his academic institution/employer and is fully compliant with NIH policies and institutional policies that is managing this potential conflict of interest; WSED is also a Senior Editor for *eLife*. The other authors declare that no competing interests exist.

### Funding

| Funder | Grant reference number | Author |
|---|---|---|
| American Association for Cancer Research | Minority Scholar Research Award | Liz Hernandez Borrero |
| National Institutes of Health | CA176289 | Wafik S El-Deiry |
| American Cancer Society | | Wafik S El-Deiry |
| Teymour Alireza P'98, P'00 Family Cancer Research Fund established by the Alireza Family | | Wafik S El-Deiry |

The funders had no role in study design, data collection and interpretation, or the decision to submit the work for publication.

### Author contributions

Liz Hernandez Borrero, Conceptualization, Data curation, Formal analysis, Validation, Investigation, Visualization, Methodology, Writing - original draft, Writing - review and editing; David T Dicker, Formal analysis, Methodology; John Santiago, Formal analysis, Visualization; Jennifer Sanders, Supervision, Writing - review and editing; Xiaobing Tian, Investigation, Methodology; Nagib Ahsan, Formal analysis, Visualization, Writing - review and editing; Avital Lev, Formal analysis, Investigation, Methodology; Lanlan Zhou, Formal analysis, Investigation, Visualization; Wafik S El-Deiry, Conceptualization, Resources, Formal analysis, Supervision, Funding acquisition, Investigation, Visualization, Methodology, Writing - original draft, Project administration, Writing - review and editing

### Author ORCIDs

Liz Hernandez Borrero https://orcid.org/0000-0003-4249-5358
Wafik S El-Deiry https://orcid.org/0000-0002-9577-8266

### Ethics

Animal experimentation: This study was performed in strict accordance with the recommendations in the Guide for the Care and Use of Laboratory Animals of the National Institutes of Health. All of

the animals were handled according to approved institutional animal care and use committee (IACUC) protocol at Fox Chase Cancer Center (protocol #14-17).

### Decision letter and Author response
Decision letter https://doi.org/10.7554/eLife.70429.sa1
Author response https://doi.org/10.7554/eLife.70429.sa2

## Additional files

### Supplementary files
- Source data 1. Gel blots.

- Supplementary file 1. Table includes gene expression values of DMSO vehicle control and analog #4 SW480 treated cells at 12 hr samples analyzed by microarray Affymetrix Human Gene 2.0 ST array probe set. Experiment included duplicates of each condition. Gene expression low expression cutoff of probe signal intensity was set at 50 (unless at least one sample did not meet this criteria for that particular probe). Normalization was performed using the RMA method and Limma eBayes for the statistical method. The present table includes values without the FDR of <0.05 filter and therefore referred as all genes. Data set applies to *Figure 2* and *Figure 2—figure supplements 1–11*.

- Supplementary file 2. Table includes gene names of DMSO vehicle control and analog #4 SW480 treated cells at 12 hr samples analyzed by microarray Affymetrix Human Gene 2.0 ST array probe set. The present table includes gene names without the FDR of <0.05 filter and therefore referred as all genes. Genes that met the FDR of <0.05 filter are referred as the differentially expressed genes (DEGs). Gene names from the data sets were compared to the Fisher Table S3 data set referred as the known p53 target gene set (*Fischer, 2017*).

- Supplementary file 3. Table includes protein information of all proteins detected in DMSO vehicle control, CB002 and analog #4 SW480 treated cells at 24 hr samples (performed in triplicates) analyzed by LC-MS/MS. The present table includes protein names and their expression values without the FDR of 1% as a filter and therefore referred as raw proteomic data.

- Transparent reporting form

### Data availability
All data generated or analysed during this study are included in the manuscript.

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
