## [Decision Letter]

**Acceptance summary:**

In this paper, a sub-group of xanthine analogues with single agent anti-tumor properties has been recognized to act through a unique mechanism of action involving significant restoration of the p53 pathway transcriptome, independently of p53, in tumors with mutated-p53 and the compounds trigger features of an S-phase checkpoint. The properties of this sub-class of xanthine compounds are different from classical xanthines such as caffeine, pentoxifylline, and theophylline that do not restore p53 pathway signaling in tumors with mutant p53 and which deregulate a G2-checkpoint rather than induce an S-phase checkpoint. The novel mechanism appears to involve transcription factors in the integrated stress response such as ATF3 and ATF4 leading to p53 and p73-independent pro-apoptotic Noxa upregulation and tumor cell death.

---

## [Author Response]

[Editors' note: we include below the reviews that the authors received from another journal, along with the authors’ responses.]

This manuscript has been reviewed by three experts in the field, and all have similar concerns with lack of clarity of some of the figures and figure legends, and more particularly with lack of clarity as the mechanism of action of some of the compounds, and the conclusion that the compounds are lethal due to p53 pathway reactivation. Unfortunately the recommendation of all three Reviewers was reject. I know this decision will be disappointing for the authors, but it is my hope that these comments are useful to them as they plan next steps; please discuss with them the possibility of addressing these concerns and possibly transferring with reviews to MCT.

We believe the manuscript previously submitted was misinterpreted by the reviewers. We never stated that the small molecules in this study activated p53, were p53 dependent, or rescued the mutant p53 to wild-type p53 activity. Instead, we claimed that these small molecules restore the p53 pathway by bypassing direct p53 activation. Our approach to discovering p53 pathway restoring compounds has over the last two decades involved live cell-based screening for functional restoration of p53-regulated reporter activity, coupled with cell death induction as a function of dose and time when we have tested chemical libraries. Thus, small molecule lead compounds and structural-analogues are not for the most part expected to act directly on mutant p53 or restore binding of mutant p53 to genes normally regulated by p53. Thus, there is no expectation that CB002 or its analogues will cause mutant p53 to bind to DNA or chromatin in the regulatory regions of p53 target genes in a manner that wild-type p53 does.

We showed that the effect of parental compound CB002 and its more potent analogue #4 IC50 values were similar in an isogenic panel of HCT116 cells (p53 null, p53 wild-type and p53 mutant), consistent with the fact that these small molecules are p53-independent. We describe the small molecules as compounds that restore the p53 signaling pathway through the involvement of other pathways. We showed that the integrated stress response ATF3 and ATF4 transcription factors play a role in the stimulation of pro-apoptotic p53 target gene, Noxa, a key mediator of apoptosis by these small molecules. Using the published Fischer p53 data-set (343 genes), we detected 197 genes and out of those, 102 were differentially expressed by analogue #4 indicating that nearly 50% of the established p53 target genes were altered.

Reviewer #1 (Reviewer Comments to the Author):In this manuscript, Hernandez Borrero et al. have extended their studies of the compound CB002 that they initially published on in Cell Cycle in 2018. In that paper they showed that this compound causes p53R175H degradation and a cell death mechanism that was dependent on Noxa. Here they attempt to elucidate this mechanism further as well as test analogues of the molecule that were synthesized in an attempt to improve potency. They show using a small number of cell lines induce Noxa independently of p73. They also show by transcriptomic and proteomic analysis that the compounds inhibit the cell cycle. Cell cycle analysis shows that compound cause an arrest in S-phase. Overall the paper has a number of major and minor problems that cause it not to be suitable for publication in the Journal. The biggest issue is the claim that the compounds restore the p53 signaling when the proteomic and transcriptomic data only weakly demonstrate that this. It s not clear if they are trying to claim that the compounds restore wild type function to mutant p53 by restoring site specific DNA binding (though it seems the compound activity is independent of p53). Another conclusion one can draw from this data is that the compounds have anti-cancer properties that kill cancer cells (more so than non-cancer cells) by an apoptotic mechanism. They also arrest cells in the Sphase of the cell cycle. Much of the transcriptomic and proteomic data are poorly annalzyed and poorly reported.The structures of compounds are missing as are the rationale and depiction of the synthesis scheme.

We thank the reviewer for their comments. We would like to clarify that the analogues tested were not synthesized by our laboratory but rather were commercially available and categorized as analogues within the commercially available Chembridge library, where the parental CB002 compound was originally obtained from. Thus, the synthesis scheme is not provided.

We never claimed that the small molecules in this study rescued the mutant p53 to wildtype p53 activity. Instead, we attest that these small molecules restore the p53 pathway by bypassing direct p53 activation.

The updated manuscript included a thorough analysis of the transcriptomic and proteomic date. At the transcriptome level, we showed that using the Fischer p53 data set (343 genes), we detected 197 genes and out of those, 102 were differentially expressed by analogue #4 indicating that nearly 50% of the established p53 target genes were altered. We discussed in the manuscript the limitations in our proteomic study including the detection of proteins. For example, DR5 and Noxa were not detected in the proteomics analysis but were validated by Western blot analysis.

Figure 1a – the p53 reporter is not sufficiently described in the results or the methods section (what is the construct of this? ). Is this done in cells? What cells?The doses of the compounds are not provided here.

Further description about the reporter, cells used and drug concentrations were added in the results and methods sections.

Figure 1b,c – again, what are the cells used here?

Further description about the cells used and drug concentrations were added in the Results section and figure legends.

Figure 1f – need to state what the p53 status is for these cells line? And need to perform cell growth inhibition on these cells lines to see if the response to some of the agents is dependent on p53 (should use WT, null, mutant +/- siRNA)

The p53 status for each cell line on Figure 1F now figure 1G has been added to the methods section. We do not claim that the effects of the small molecules is dependent on direct p53 activation. In the cell lines tested, restoration of the p53 pathway by the small molecules is independent of p53, thus it is not expected that Knockdown of p53 will have an effect. Moreover, our experiments in an isogenic panel of HCT116, including p53 null cells suggests that the mechanism of action is independent of p53. The idea that restoration of the p53 transcriptome by pathways involving alternative transcription factors and the extent to which we have demonstrated this in our paper we believe is at the forefront of the field.

Figure 1g – belongs in the supplemental.

We thank the reviewer for this comment nonetheless, we believe this key finding pertains as a main figure. The manuscript has been re-arranged and original figure 1G is now shown as figure 1H.

Figure 2A – Again, it is striking that the cell line (and p53 status) used here is not mentioned in the Results section or the figure legend. there are only 6 p53 targets shown here (there should be many more). Further, the increase in their expression is only mild.

Figure 2A is now figure 2A-B. Microarray data has been reanalyzed. Analysis of p53-responsive genes that were differentially expressed by the analogue #4 showed that, using the Fischer p53 data set (343 genes), we detected 197 genes and out of those, 102 were differentially expressed by analogue #4 indicating that nearly 50% of the established p53 target genes were altered. This is a significant overlap of p53 regulated genes. Details regarding the cell line used for transcriptomics are located in the figure legend and result section. P53 status for cell line used are located in the materials and method section.

Use of the term "reactome" is confusing. In the Results section they state they use both transriptomic and proteomic data, so which are "reactome" are they both?

The term reactome was edited to either transcriptome or proteome when applied to be more clear on what is meant.

Figure 2A the data showed in the pathway analysis of genes upregulated and down regulated is not discussed in the Results section. Nor really is the proteomic data in Figure 2B. The only mention is that the cell cycle pathway is down. Again, this data is very poorly explained. I do not see how this really supports the conclusion that mechanism of cell growth inhibition is through the p53 pathway.

The points about the role of the p53 protein versus the p53 pathway are explained more in the response to reviewers and in the manuscript. The manuscript has been re-arranged and Figure 2A and 2B are now Figure 2 and Figure 3, respectively. The transcriptomic data has been re-analyzed. The discussion from the results of the transcriptomic and proteomic analysis has been extended and incorporated into the main text of the Results section (now also including Figure 4). We have highlighted in the manuscript the restoration of the p53 pathway at the transcriptional level. Our discussion includes limitations of the results observed at the transcriptional and protein level. For example, DR5 and Noxa were not detected in the proteomic analysis but were validated by Western blot analysis.

Figure S5 – the venn diagrams show actually very few p53 target proteins in common when looking at jus the in house database or the published database, and even fewer that are in common with all three. This is not strong data to support their conclusion.

We thank the reviewer for this comment. We have edited the manuscript to highlight the observed restoration of the p53 pathway at the transcriptional level. Our discussion includes limitations on the results observed at the protein level. Of note, differences in protein overlaps are influenced by the type of cell line and treatment. In house proteomics was performed in HCTT116 cell lines with wild-type p53 versus null for p53. Upon treatment with 5-FU we compared our profiles with the known p53 targets from the Fischer database which is a compilation of transcriptomic data from a variety of studies using different cell lines and treatments.

Figure 3A – this blot is very hard to follow. CB002 some of the data appear to contradict their statements in the results for example they say Lines 195, 196 – "Similarly, the combination of etoposide with CB002 or CB002-analogue #4 showed a decrease in expression of pcdc2(Tyr15) and pcdc25c(Ser16)." However the blot shows that the pcdc2(tyr15) levels increase in combo with etoposide.

We have re-run the western blot from Figure 3A now figure 5A to improve the clarity of the result. Both proteins, pcdc2(Tyr15) and pcdc25c(Ser16) decrease in single treatments compared to DMSO control. A slight increase is observed after small molecule treatment with etoposide as compared to single small molecule treatment we believe due to an etoposide effect on G2/M.

Figure 3B – I assume that this was done using flow cytometry but that is not stated.

Details from figure 3B now figure 5E, in addition to be included in the methods section, have been added to in the Results section and figure legends.

Figure 3E – including 7 time points is excessive when the data can be communicated with three and sometimes just two time points. It makes the figures very difficult to read.

We thank the reviewer for this comment. Figure 3E is now figure 5E. We kindly disagree with this comment, showing the effects of the drug with just two time point would not capture how the cells are passing through the cell cycle. Therefore, to clearly show the effect of drug treatments in the cell cycle our data includes several time points from these very carefully done experiments.

Figure 4B – there does appear to be a almost two fold increase in toxicity in the WI38 cells. This is an increase. (it may not be statistically significant) but that not how it is written in the manuscript. Furthrmore, drug concentrations on the organoid in 4C go up to 100 uM, here they only go up 25 μm in the Wi38 cells. They would need to show the data in doses at 50 and 100 uM.

We thank the reviewer for this comment. Figure 4B is now figure 6B. We have edited the result section to specifically mention that the increase in Sub-G1 cell content in the WI38 cell line is not statistically significant. The treatment with CB002-analogue #4 in this experiment was performed at 25uM since this is the IC50 concentration, which was also used throughout the experiments, unless otherwise indicated.

Figure 4D should be quantified in addition to the micrographs.

Quantification of Figure 4D, now figure 6D has been included in the manuscript.

Figure 4G – the tumor growth inhibition is seen at only one time point and is mild.

We thank the reviewer for this comment. Figure 4G is now figure 6G in the updated manuscript. The current manuscript specifically states that at 5 weeks the differences between control and treated tumors are statistically significant. We have also included in our discussion that the effects are suboptimal and warrant optimization.

Reviewer #2 (Reviewer Comments to the Author):The manuscript by Hernandez Borrero et al. describes the characterization of a set of small molecules derived from a previously identified lead compound CB002 that the authors described as p53 pathway restoring compound. The new derivatives are more potent in inducing cell cycle disruption and inducing cell death in cell culture and tumor spheroid culture. The authors suggest this class compounds may be useful as novel cancer therapeutics based on their ability to restore p53 activity in tumor cells expressing mutant p53.Identifying new compounds that restore the transcriptional function of mutant p53 is a challenging but attractive strategy of targeting mutant p53 in cancer. Currently there is a set of basic criteria that true mutant p53 rescue compounds are expected to meet, such as restoring mutant p53 binding to consensus DNA sequence in vivo and activating representative wt p53 target genes. Knockout of endogenous mutant p53 should eliminate the effect, and reintroduction of mutant p53 to a p53-null cell should restore the effect. The rescue compound is likely to interact with mutant p53 protein either non-covalently or covalently to change its conformation and DNA binding activity.The compounds described in this study have not been rigorously tested in this regard and do not fit any of the above criteria. They induce a small number of genes that overlap with p53responsive genes, but it is unclear whether the induction is mediated by mutant p53. But curiously strong wt p53 target genes p21, MDM2, PUMA etc are not induced by these compounds. The results do not rule out the possibility that the small overlap in gene expression profile is coincidental, since p53 target genes can be regulated by other pathways and factors. Based on the data presented, it is premature to call these compounds mutant p53-rescue compounds or p53-pathway activating drugs.The data mainly focused on analyzing the cell cycle and toxicity profile of the compounds, which is not very informative in terms of drug development. One would have like to see evidence of how the compounds restore mutant p53 function, whether they interact with p53 protein directly, if they do not act on p53 directly, what is their cellular target. Given the absence of these information, the manuscript is not suitable for publication.

We thank the reviewer for their comments. We would like to clarify that we did not state in the manuscript that the small molecules activate p53, are p53-dependent, or rescue the mutant p53 to wild-type p53 activity. Rather, we attest that these small molecules restore the p53 pathway by bypassing direct p53 activation. Thus, it is not expected that CB002 or its analogues will cause mutant p53 to bind to DNA or chromatin in the regulatory regions of p53 target genes in a manner that wild-type p53 does. Therefore, the small molecules are not mutant p53-rescue compounds but rather p53-pathway restoring or bypass compounds.

We performed a microarray analysis to identify p53-responsive genes that were differentially expressed by the treatment of the most potent analogue #4. At the transcriptomic level, we showed that using the Fischer p53 data set (343 genes), we detected 197 genes and out of those, 102 were differentially expressed by analogue #4 indicating that nearly 50% of the established p53 target genes were altered. This is a significant overlap of p53-regulated genes. We validated several p53 target genes including DR5 and Noxa. In the original manuscript p21 was not shown but we extended this observation in the updated manuscript (Figure 5D). MDM2 and Puma were not shown therefore it is not accurate to state that these target genes were not induced, rather they were not part of the scope of the shown experiment.

Although the molecular target was not identified in this study, we provide evidence that the integrated stress response ATF3 and ATF4 transcription factors play a role in the stimulation of pro-apoptotic p53 target gene, Noxa, a key mediator of apoptosis by these small molecules.

Some specific commentsFigure 1A. compound concentration range in the p53 reporter assay was not provided. Is mutant p53 expression needed to observe the activation of p53 reporter? Does crispr knockout of mutant p53 abrogate the reporter activation by the compounds? #4 has strong cell kill but modest p53 reporter activation.

Compound concentration ranges were added to the results and methods sections. Mutant p53 is not required to activate the p53 reporter thus a CRISPR knockout would not be expected to abrogate the p53 reporter activity. At the conditions tested (6 hr) for the p53 reporter assay, CB002 analogue #4 had a modest reporter activation but showed the strongest cell killing in the cytotoxicity assays performed at 48 hrs. Because the p53 reporter assay is based on early transcriptional effects, we believed that at 6 hrs the signaling effect for analogue #4 may have already ceased given that it was the most potent drug and most likely the signal transduction was initiated and terminated compared to the other compounds tested.

Fig1F. Text mentioned HCT116 and HCT116-p53- with exogenous p53 mutant, but data does not contain these cells.

Figure 1F now Figure 1G now includes data from HCT116 and HCT116-p53- with exogenous p53 R275H mutant.

FigS5 Compound 4 induced genes only have very few overlap with known p53 targets. Does not support the notion that the compound activates p53 pathway, possibly coincidental overlap.

We thank the reviewer for this comment. FigS5 has been updated to Figure 2A due the microarray re-analysis. Analysis of p53-responsive genes that were differentially expressed by CB002 analogue #4 showed that, using the Fischer p53 data set (343 genes), we detected 197 genes and out of those, 102 were differentially expressed by analogue #4 indicating that nearly 50% of the established p53 target genes were altered. This is a significant overlap of p53 regulated genes.

Reviewer #3 (Reviewer Comments to the Author):The manuscript by Hernández Borrero et al. describes a small molecule, CB002, that is said to activate the p53 pathway in cancers with mutant p53. The authors observed that the compound leads to S-phase checkpoint activation, with increased phosphorylation of RPA, ATR, cyclin E and decreased levels of phosphorylated H3 and proteins related to DNA synthesis. Moreover, the authors report that CB002 and its analogues induce apoptosis in cells, as seen through higher proportion of sub-G1 cells, increased PARP cleavage and a release of cytochrome c from the mitochondria. Noxa was shown to be upregulated upon treatment with CB002 and its analogues. Importantly, Noxa depletion inhibited the anti-tumor effects exerted by analogue #4 in vivo, suggesting an important role of Noxa in mediating apoptosis downstream of this compound. The compound tested showed limited toxicity in normal cells, as shown in cell culture and mouse studies, further suggesting a potential of this compound to be brought forward for further clinical studies. The authors propose this molecule as a potential therapeutic approach in targeting tumors that have a p53 mutation.The potential of CB002 and its derivatives can be appreciated from this manuscript. However, the currently reported data do not provide evidence that these compounds act through p53. Without thorough additional experimentation, it cannot be concluded that p53 reactivation is the major mechanism by that these drugs are acting. As it stands, it is at least equally possible (if not more conceivable) that these are cytotoxic drugs that act independent of p53.Overall, the manuscript text itself was well written, and easy to read. However, the figures were slightly messy and not very intuitive. Often, figure legends lack sufficient details that help readers understand the figures. Certain labels are cropped off and not aligned very well with the results. The use of many different CB002 analogues is appreciated, but the lack of consistency between the analogues shown in the results is also confusing. The manuscript would benefit from mentioning why a particular analogue (and not the others) was chosen and shown for a particular experiment.

We thank the reviewer for their comments. We would like to clarify that we did not state in the manuscript that the small molecules activate p53, are p53-dependent, or rescue the mutant p53 to wild-type p53 activity or conformation. Rather, we claim that the mechanism of these drugs involves the restoration of the p53 pathway by bypassing direct p53 activation. We provide evidence that the integrated stress response ATF3 and ATF4 transcription factors play a role in the stimulation of pro-apoptotic p53 target gene, Noxa, a key mediator of apoptosis by these small molecules.

CB002 analogues were prioritized by their ability to induce Noxa expression as shown in Figure 1G. We then show that a subset of these analogues in Figure 1C induce apoptosis. As we started studying the cell cycle effects, we focused on the analogues that fit the criteria of the new class of small molecules i.e. perturbation of the S-phase of the cell cycle and restoration of the p53 pathway. We then concentrated the majority of the manuscript on the most potent CB002 analogue #4.

Major comments:1. Although the clinical potential of these compounds is interesting, especially considering its lack of toxicity in normal cells, the link to p53 is weak. As the authors are presenting these compounds as "activators" of p53, this link would need to be strengthened, unless the authors re-consider their concept and describe CB002 and its derivatives as cytotoxic drugs with as yet unknown mechanisms of action.

We appreciate these comments. We respectfully disagree with the reviewer and would like to clarify that we did not state in the manuscript that the small molecules activate p53. Rather, we claim that the mechanism of these drugs involves the restoration of the p53 pathway by bypassing direct p53 activation. We have edited the manuscript to highlight the restoration of the p53 pathway at the transcriptional level. We showed that using the Fischer p53 data set (343 genes), we detected 197 genes and out of those, 102 were differentially expressed by analogue #4 indicating that nearly 50% of the established p53 target genes were altered.

It seems unclear if the compounds activate p53 as the major mechanism to induce apoptosis, S-phase arrest, decreased proliferation (Ki67) and therefore reduction in tumor growth. Testing for p53 reporter activity is not sufficient. It would be important to observe the entire set of common p53-responsive genes (e.g. the CDKN1A gene discovered to be p53-responsive by the senior author, or Mdm2, Puma, etc., perhaps referring to the catalogues presented by M. Fischer, doi: 10.1038/onc.2016.502, upon addition of these compounds).

We appreciate the reviewer’s comment. We wish to clarify that we do not claim that the mechanism of these drugs involves the activation of p53 for the restoration of the p53 pathway. Rather they restore the p53 pathway through bypassing the direct activation of p53 via other molecular pathways. We provide evidence that the integrated stress response ATF3 and ATF4 transcription factors play a role in the stimulation of pro-apoptotic p53 target gene, Noxa, a key mediator of apoptosis by these small molecules. We do not conclude p53 pathway restoration solely on the p53-responsive luciferase reporter which was a preliminary result years ago at an early point in the history of this work. We incorporate a microarray analysis and compare the differentially expressed genes to the Fischer data set. In this comparison, we detected 197 p53-responsive genes and out of those, 102 were differentially expressed by CB002 analogue #4 indicating that nearly 50% of the established p53 target genes were altered. We have also validated some of these targets at the protein level by western blot analysis showing increased expression of p53 target genes p21, Noxa, and DR5.

Adding on to this point, it would also be important to prove that the induction of Noxa was through p53 activation. Knocking down p53, and testing for Noxa expression upon treatment with these compounds would be essential to make this claim.

We wish to clarify that we do not claim that the mechanism of these drugs involves the activation of p53 for the restoration of the p53 pathway. Restoration of the p53 pathway in mutant p53 harboring cells by these small molecules is independent of p53, thus it is not expected that knockdown of p53 will have an effect. Moreover, our experiments in an isogenic panel of HCT116, including p53-null cells suggests that the mechanism of action is independent of p53.

I find it puzzling that the transcriptome data comparison with in-house p53 proteomic database and known p53 targets had such little overlap. Does this not further suggest that the compounds are acting through a different pathway, independent of p53?

We thank the reviewer for this comment. We have highlighted in the manuscript the restoration of the p53 pathway at the transcriptional level. Our discussion includes limitations in the results observed at the transcriptional and protein level. For example, DR5 and Noxa were not detected in the proteomics analysis but were validated by Western blot analysis.

We have also included in our discussion how a large number of differentially expressed genes involved in the cell cycle can be connected to p53 pathway restoration. The p53 target p21 is involved in the downregulation of cell cycle genes through its association with the DREAM complex, consequently E2F gene targets become downregulated. This connection was suggested as a future direction and was beyond the scope of this manuscript.

According to Supp. Figure 10C, kd of ATF3/4 completely abolishes Noxa induction. This can be due to one of two reasons; the compounds activate p53 which leads to induction of ATF3/4, then to Noxa, or these compounds lead to the activation of other pathways that induces ATF3/4 (i.e: the Integrated stress response). Therefore, the p53 knockdown experiment is important here to prove the role of p53 in their system. On the same note, I am curious to know if these cells in Supp. Figure 10C with ATF3/4 kd survive CB002/ analogue #4 treatment. If Noxa is the key mediator of cell death here, one would expect that these cells escape the apoptosis-inducing effects of the compounds.

Supp Figure 10C is now Figure 1I. We thank the reviewer for this comment. Restoration of the p53 pathway in mutant p53-harboring cells by these small molecules is independent of p53, thus it is no expected that knockdown of p53 will have an effect on Noxa expression. Studies from our lab suggest novel involvement of integrated stress response transcription factors ATF3/4 in the induction of p53 target genes (10.1016/j.neo.2021.01.004) and this manuscript suggests similar based on its findings with a different class of compounds that bypass the p53 pathway.

2. Analogue #4 appears to exert its toxicity in tumor cells irrespective of the p53 status (Figure 4A) – roughly the same IC50 in HCT116 p53-/- or p53 +/+ or the R273H mutant.Could the authors explain how the compound exerts its effects in the p53 null cell line, and why these cells would have similar IC50s for this analogue? Does this not further suggest an alternative pathway of activation downstream of the compound? The integrated stress response again was mentioned in the discussion (line 301). It would be of high interests to test if these compounds lead to apoptosis or an S-phase checkpoint activation because of the stress response rather than p53. This would provide an entirely different explanation for the observations presented in the rest of the paper.

We thank the reviewer for this comment. We want to clarify that the restoration of the p53 pathway by these compounds in mutant p53-harboring cells is independent of p53. Along these lines, our experiments in an isogenic panel of HCT116, including p53-null cells suggest that the mechanism of action is independent of p53.

We showed that the integrated stress response ATF3 and ATF4 transcription factors play a role in the stimulation of pro-apoptotic p53 target gene, Noxa, a key mediator of apoptosis by these small molecules. In regards to the S-phase activation, we have also included in our discussion how a large number of differentially expressed genes involved in cell cycle control can be connected to the p53 pathway restoration. The p53 target p21 is involved in the downregulation of cell cycle genes through its association with the DREAM complex, consequently E2F gene targets become downregulated. ATF3/4 have been implicated in the regulation of p21. This connection was suggested as a future direction and was beyond the scope of this manuscript.

Other comments:1. Could the authors test a non-53 reporter construct as a negative control in the experiment in Figure 1A.

We appreciate the reviewer’s comment. The specificity of the reporter used in this study has been validated in the original PNAS paper published in 2006 from our laboratory including the changes at endogenous targets including p21, DR5, Noxa (as well as in the original paper of the p21 (WAF1) discovery by El-Deiry et al., Cell, 1993). We understand that reporters can be misleading and we thus validated endogenous targets in this study including p21, DR5, and Noxa which is really very important.

2. I am not too convinced just by the pictures in Figure 1E. Could the authors show the different channels separately as well? Would it be possible to provide the quantification of the green intensity in the IF pictures?

We have enhanced the image to improve clarity of the result.

3. The authors described on the manuscript text (line 135 – 137) that they observed Noxa induction upon treatment with the CB002 analogues in DLD-1, SW480, HCT116, and HCT116 p53-/- tumor cells expressing the exogenous R175H p53 mutant. However, the figure only shows Noxa expression in DLD-1 and SW480 cells.

Figure 1F now Figure 1G includes data from HCT116 and HCT116-p53- with exogenous p53 R275H mutant.

4. The heatmap of Figure 2B confused me. According to the legends, the authors treated the cells with analogue #4 (T4) for 12 hours but CB2 for 24 hours. Are they comparing these 2 treatments of different timepoints here?

The figure legend has been edited for clarification. Transcriptomics with analogue #4 were done at 12 hrs, proteomics for both CB002 and analogue #4 were performed at 24 hrs.

5. Could the authors include the total levels of H3 and total cdc25 in Figure 3A and also the total levels of RPA and ATR in Figure 3B?

Protein totals have been added to the requested proteins in the western blot that is shown.

6. The gating used for the flow cytometry (as shown in Figure 3E and Supp. Figure 9) does not appear the same across the different samples. This would of course lead to bias in analysis. Please check and re-gate/analyze if necessary.

We appreciate the reviewer’s comment. Figures 3E and supplemental Figure 9, now Figure 5F and S14, respectively were independent experiments and compared to each negative control. Gating was maintained constant on each set of experiments and gates were set as per the negative control for each one.

7. The labels in the figures: Please change the μm to μM.

The symbols for μM have been corrected.